# Galectin-3 is required for the microglia-mediated brain inflammation in a model of Huntington's disease

Jian Jing Siew [1,2], Hui-Mei Chen[2], Huan-Yuan Chen[2], Hung-Lin Chen [2], Chiung-Mei Chen[3], Bing-Wen Soong [4,5], Yih-Ru Wu[3], Ching-Pang Chang [2], Yi-Chen Chan[6], Chun-Hung Lin[6], Fu-Tong Liu [1,2] & Yijuang Chern [1,2]

Huntington's disease (HD) is a neurodegenerative disorder that manifests with movement dysfunction. The expression of mutant Huntingtin (mHTT) disrupts the functions of brain cells. Galectin-3 (Gal3) is a lectin that has not been extensively explored in brain diseases. Herein, we showed that the plasma Gal3 levels of HD patients and mice correlated with disease severity. Moreover, brain Gal3 levels were higher in patients and mice with HD than those in controls. The up-regulation of Gal3 in HD mice occurred before motor impairment, and its level remained high in microglia throughout disease progression. The cell-autonomous up-regulated Gal3 formed puncta in damaged lysosomes and contributed to inflammation through NFκB- and NLRP3 inflammasome-dependent pathways. Knockdown of Gal3 suppressed inflammation, reduced mHTT aggregation, restored neuronal DARPP32 levels, ameliorated motor dysfunction, and increased survival in HD mice. Thus, suppression of Gal3 ameliorates microglia-mediated pathogenesis, which suggests that Gal3 is a novel druggable target for HD.

[1] Taiwan International Graduate Program in Molecular Medicine, National Yang-Ming University and Academia Sinica, Taipei 11529, Taiwan. [2] Institute of Biomedical Sciences, Academia Sinica, Taipei 11529, Taiwan. [3] Department of Neurology, Chang Gung Memorial Hospital, Linkou Medical Center and College of Medicine, Chang-Gung University, Taoyuan 33302, Taiwan. [4] Department of Neurology, Shuang Ho Hospital, and Taipei Neuroscience Institute, Taipei Medical University, Taipei 23561, Taiwan. [5] Department of Neurology, Taipei Veterans General Hospital, and Brain Research Center, National Yang-Ming University, Taipei 11221, Taiwan. [6] Institute of Biological Chemistry, Academia Sinica, Taipei 11529, Taiwan. Correspondence and requests for materials should be addressed to Y.C. (email: bmychern@ibms.sinica.edu.tw)

Huntington's disease (HD) is an autosomal dominant degenerative disease that is caused by the expansion of CAG repeats in the *Huntingtin* (*HTT*) gene[1]. When the number of CAG repeats exceeds 36, the translated polyglutamine (polyQ)-containing HTT protein (mHTT) forms inclusions, subsequently jeopardizing important cellular machinery in various types of brain cells (e.g., neurons, astrocytes, and microglia)[2]. Clinical presentations of HD include an impairment of motor and cognitive functions, brain atrophy, body weight loss, and shortened lifespan[3,4]. Multiple HD animal models, ranging from *Caenorhabditis elegans* to monkeys, have been used to study the pathogenic mechanisms and potential treatments[5]. Despite the tremendous efforts devoted to the development of therapeutic interventions, there is currently no cure or disease-modifying drugs available for this devastating disease. Moreover, appropriate biomarkers have yet to be defined for accurate and sensitive evaluation of HD progression[6,7].

Chronic inflammation has been implicated in the pathogenesis of many neurological diseases[8–10]. In the brain, the major cell type responsible for inflammatory responses is microglia. Many noxious factors (including cytokines, reactive oxygen species, and nitric oxide, NO) are released by microglia and cause widespread neuropathology[11]. Previous studies have suggested that the aberrant activation of microglia contributes to HD pathogenesis. For example, autonomous microglia activation in the presence of mHTT instigates sterile inflammation[12]. Abnormal activation of nuclear factor (NF)κB-p65 in microglia and astrocytes causes enhanced production of pro-inflammatory factors during HD progression[13]. Most importantly, therapeutic interventions aimed at thwarting neuronal inflammation impede disease progression[14].

Galectins (Gals) are a family of lectins, which contain a conserved carbohydrate-recognition domains for β-galactosides[15]. Besides interacting with glycoproteins, Gals also have non-carbohydrate-binding partners and may function intracellularly. There are a total of 15 Gals and are categorized into three groups (i.e., proto, tandem repeats, and chimera). Galectin-3 (Gal3) is the only member in the chimera group, and is expressed in humans and mice[15]. Gal3 contains no classical leader signal, and can be found in various cellular compartments including the nucleus, cytoplasm, plasma membrane, and in the extracellular space[16]. Although up-regulation of Gal3 has previously been reported in microglia[16,17], the role of Gal3 in neuroinflammation is not fully understood. In experimental autoimmune encephalomyelitis, deficiency of Gal3 reduces inflammation and disease severity[18]. In transient focal brain ischemia, expression of Gal3 is associated with neuronal death[17]. Conversely, reduction of Gal3 exacerbated the inflammatory response in a mouse model of amyotrophic lateral sclerosis, in which Gal3 was up-regulated in the spinal cord[19]. These findings suggest that Gal3 might serve as a key player that controls the switch between the detrimental and protective effects of microglia. Since Gal3 is present in various subcellular locations and can be modulated by multiple modifications (such as serine phosphorylation[20] and oligomerization[21]), it is not surprising that Gal3 exhibits multiple and/or seemingly opposing functions under various physiological conditions. Further characterization of the role of Gal3 in neurodegenerative diseases is a timely issue because it has been documented that aggregations of several disease-causing proteins (such as α-synuclein, Tau, and Huntingtin) trigger rupture of intracellular vesicles, which are marked by the appearance of Gal3 puncta, in cell models of degenerative diseases[22]. This is of particular interest because impaired lysosomes and autophagosomes in HD have been reported[23]. In the present study, we found that Gal3 is up-regulated in the brains of HD patients and two HD mouse models (R6/2, HdH[150Q]). Gal3 puncta are detected in lysosomes of HD microglia. Suppressing Gal3 by genetic and pharmacological approaches reveal that up-regulated Gal3 levels in microglia contributes to HD pathogenesis in a feed-forward NFκB-dependent and NLRP-3 inflammasome-dependent manner. Collectively, our findings provide mechanistic insights into the role of Gal3 in HD and suggest Gal3 as a potent target for therapeutic intervention for HD.

## Results

**Gal3 up-regulation in the brain and plasma of HD patients.** Previous studies have reported that the levels of plasma or serum Gal3 in patients with several neurological diseases were higher than those in control individuals[24–26]. We sought to evaluate the plasma Gal3 levels of HD patients (Supplementary Table 1). In a pilot study, ELISA analysis revealed that HD patients ($n = 26$) had a higher level of plasma Gal3 than did pre-symptomatic HD ($n = 4$) and non-HD individuals ($n = 16$) (Fig. 1a). Most importantly, the levels of plasma Gal3 correlated well with disease burden (Fig. 1b), Mini-Mental Status Examination (MMSE) scores (Fig. 1c), and Unified Huntington's disease Rating Scale (UHDRS) scores (UHDRS-Motor, Fig. 1d; UHDRS-Independence, Fig. 1e; UHDRS-Functional Capacity, Fig. 1f).

To evaluate whether the amount of Gal3 in the brain of patients with HD was also higher than that in non-HD controls, we analyzed the expression of Gal3 transcripts (i.e., *LGALS3*) in post-mortem caudate putamen and cerebellum samples of HD patients and non-HD subjects (Fig. 1g, h, Supplementary Table 2). Quantitative reverse transcription PCR (RT-qPCR) analysis showed significant up-regulation of *LGALS3* transcripts in the caudate putamen, the most affected brain area[27] of HD patients ($n = 5$ in each group). Although a trend toward Gal3 up-regulation was observed in the cerebellum of HD patients, no statistically significant difference was found ($P = 0.09$; Fig. 1h), probably due to a marked variation among HD cerebellar specimens. Previous studies showed seemingly contradictory observations on the cerebellar functions of HD patients[28,29], indicating that the involvement of cerebellum in HD remains elusive. We thus chose the caudate nucleus (striatum) for further investigation. We performed immunofluorescence staining and found increased levels of Gal3 in microglia of the caudate putamen of HD patients (Supplementary Fig. 1, Supplementary Table 3). Collectively, our findings suggest that HD patients have increased Gal3 levels in the plasma and caudate putamen.

**Up-regulation of Gal3 in HD mouse models.** We evaluated whether the levels of Gal3 were up-regulated in the plasma of R6/2 mice, an HD mouse model that harbors exon-1 of the human mHTT gene[30]. An ELISA showed significant up-regulation of plasma Gal3 at the disease-manifested stage (12 weeks old) but not earlier (Fig. 2a). The levels of Gal3 in the striatum of R6/2 mice were also analyzed. RT-qPCR analysis showed that up-regulation of Gal3 was detected in the striatum at the age of 7 weeks, when motor dysfunction had not yet been observed, and remained at a high level as the disease progressed to the end stage (12 weeks old; Fig. 2b). Consistent with this finding, increased Gal3 protein levels were found in the striatum of R6/2 mice at 12 weeks compared with those of WT mice (Fig. 2c). Immunofluorescence staining showed a significant increase in Gal3 protein levels in the striatum of R6/2 mice. Most importantly, all Gal3-positive signals were detected in IbaI-positive cells, which is indicative of microglia, but not in astrocytes or neurons (Fig. 2d, e, Supplementary Fig. 2a, b).

Moreover, primary microglia derived from R6/2 mice also had higher levels of Gal3 protein than WT microglia, as assessed by immunofluorescence staining and flow cytometry (Fig. 2f–h). The

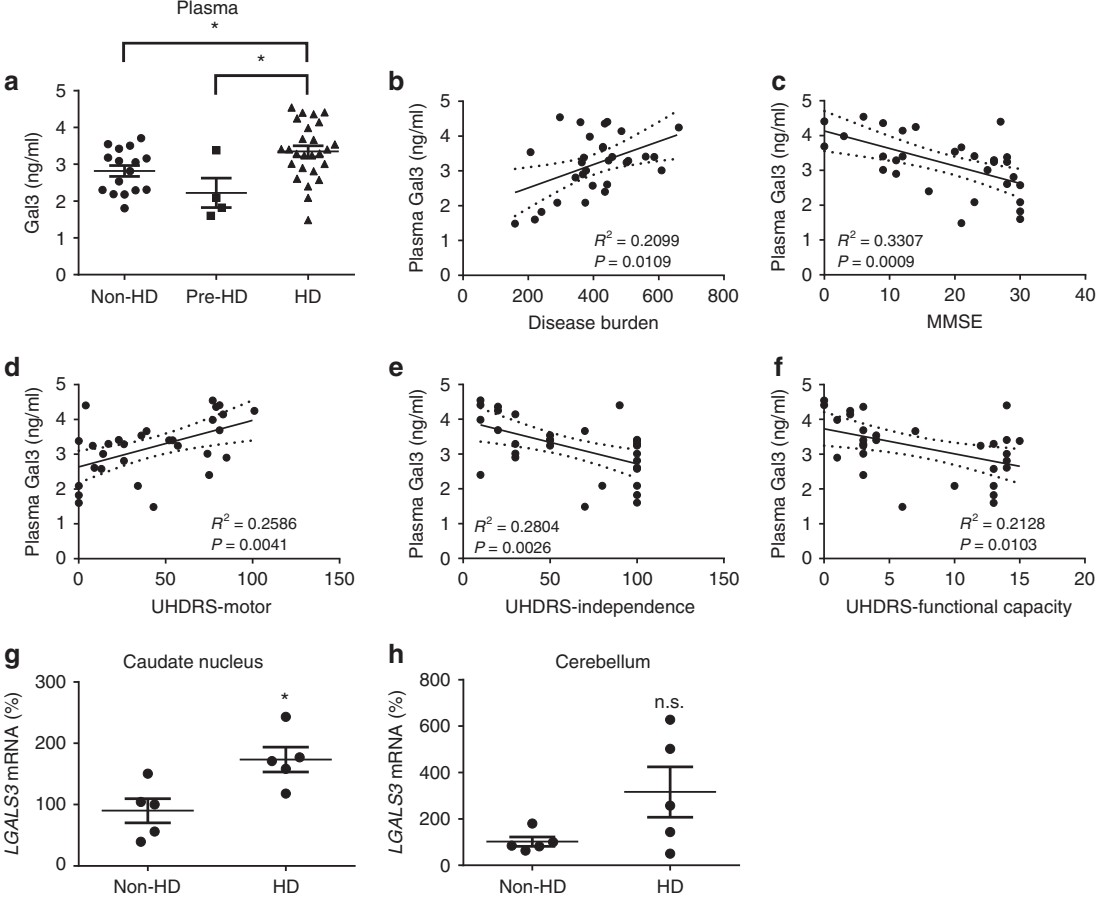

**Fig. 1** Up-regulation of Gal3 in the plasma and brains of HD patients. **a** The levels of plasma Gal3 of non-HD ($n = 16$), pre-symptomatic HD (Pre-HD, $n = 4$), and HD individuals ($n = 26$) were measured by ELISA. **b–f** Correlations between the plasma Gal3 levels of HD patients ($n = 26$) and their disease burden **b**, MMSE **c**, UHDRS-motor **d**, UHDRS-Independence **e**, or UHDRS-functional capacity **f** scores were analyzed by Pearson correlation coefficients. The levels of transcripts of *LGALS3* in the **g** caudate nucleus and **h** cerebellum of HD patients or non-HD subjects were measured by RT-qPCR ($n = 5$ in each group). Each dot represents the plasma Gal3 level or the relative *LGALS3* level of a subject. Data in (**a**) were analyzed by one-way ANOVA followed by Holm–Sidak's multiple comparisons test, $*P < 0.05$. Data in **g** and **h** are presented as the means ± SEM and were analyzed by the unpaired Student's *t*-test. $*P < 0.05$. n.s., not significant. Source data is available as a Source Data File

median fluorescence intensity of Gal3 in WT and R6/2 cells was $4628 \pm 560$ and $7749 \pm 332$, respectively (mean ± SEM, Student's *t*-test, $P = 0.0087$). *Lgals3* transcript levels were higher in R6/2 microglia as determined by RT-qPCR (Fig. 2i). Of note, the morphology and size of both R6/2 and WT microglia varied within each group. The mean size of R6/2 microglia was larger than that of WT microglia as determined by the IbaI-positive area ($100 \pm 4\%$ ($n = 297$) and $127 \pm 5\%$ ($n = 326$) for WT and R6/2 microglia, respectively; mean ± SEM, Student's *t*-test, $P < 0.0001$). To rule out the possibility that WT microglia may take a longer period of time to acquire a similar phenotype as R6/2 microglia, we performed a comparative experiment using microglia on day 14 in vitro (DIV14) and DIV21. Our data show that WT microglia at both DIV14 and DIV21 contained less Gal3 than HD microglia at DIV14 or DIV21 (Supplementary Fig. 3).

We next evaluated the level of Gal3 in a knock-in model, Hdh[150Q], at the manifested stage. Immunofluorescence staining (Fig. 2j) and qPCR (Fig. 2k) revealed significant increases in the protein and transcript levels of Gal3, as were observed in R6/2 (Fig. 2j, k). Similarly, Gal3 was detected in microglia, but not in astrocytes or neurons (Supplementary Fig. 2c, d).

**NFκB mediates the up-regulation of Gal3 in HD microglia.** We previously demonstrated that mHTT activated the NFκB pathway

and triggered inflammatory responses[14]. Because NFκB has been implicated in the regulation of Gal3 in several peripheral tissues and cells[31,32], we hypothesized that mHTT-mediated activation of NFκB might contribute to Gal3 up-regulation in HD microglia. Treatment of primary microglia with two different NFκB inhibitors, BAY11-702 (designated Bay11, 3 μM) or Ro 106-9920 (designated Ro106, 1 μM, Supplementary Fig. 4) for 24 h suppressed the activation of NFκB-p65 in nuclei and reduced the expression of Gal3 in HD microglia (Fig. 3a, b). To measure the activity of NFκB, we performed an NFκB-p65 transcription factor assay using nuclear extracts prepared from HD and WT microglia. Treatment with Bay11 suppressed NFκB-p65 activation in HD microglia (Fig. 3c). In addition, western blot analyses revealed that the elevated levels of phosphor-p65 and p65 in the nuclear fractions of HD microglia cells were reduced by Bay11 (Supplementary Fig. 5a). These results suggest that NFκB plays a central role in Gal3 up-regulation in HD microglia. In addition, inhibition of NFκB-p65 also markedly reduced the release of inflammatory cytokines (e.g., IL1β, IL6, and TNFα) and increased the production of the anti-inflammatory cytokine (IL10, as determined by ELISA (Fig. 3d). We also determined the NO levels in the medium using the Griess reagent, and we showed that Bay11 treatment significantly decreased the production of NO by HD microglia. Only a very low level of NO was found in the medium from WT microglia and that was not affected by

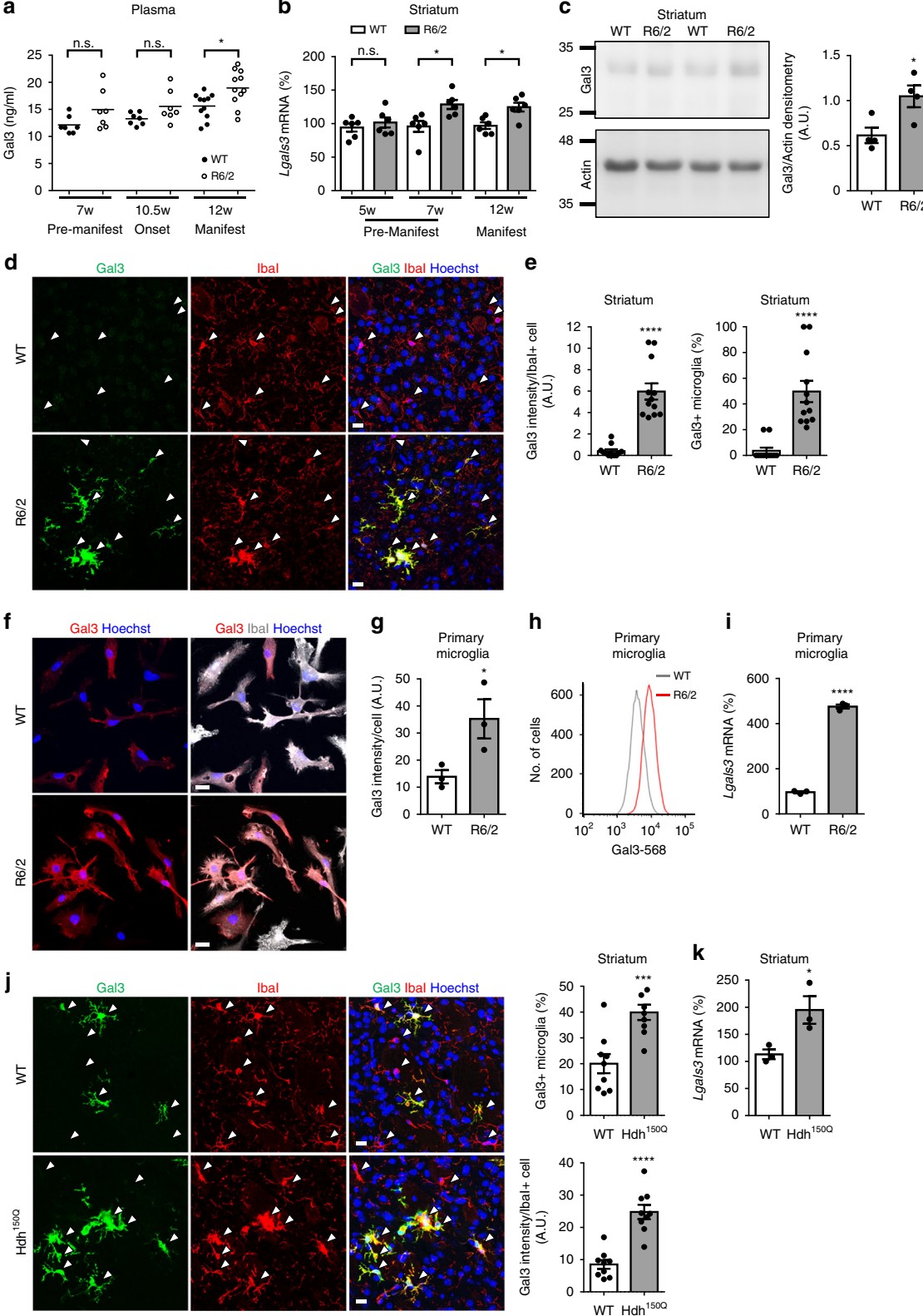

treatment with Bay11 (Fig. 3e). These data suggest that NFκB is involved in the up-regulation of Gal3 and inflammation in HD microglia.

**Gal3 promotes inflammation in primary microglia**. As Gal3 up-regulation was associated with the inflammatory response in HD microglia (Fig. 3), we next evaluated whether Gal3 was required

for the inflammatory response in HD microglia. The up-regulation of Gal3 in HD microglia was significantly reduced after 1 week of treatment with sh*Lgals3* when compared to that of HD microglia infected with a control virus harboring sh*GFP*. In addition, the nuclear level of NFκB-p65 was also decreased in the Gal3-knockdown microglia (Fig. 4a, b) suggesting that Gal3 up-regulation is important for the abnormal activation of NFκB in

**Fig. 2** Up-regulation of Gal3 in HD microglia. **a** An ELISA of Gal3 was performed on plasma collected from 7-week-old, 10.5-week-old, and 12-week-old R6/2 mice and their controls ($n = 7$–11 in each group). Each dot represents the value of an individual animal. **b** qRT-PCR analysis of *Lgals3* was performed on cDNA prepared from striatal tissues collected from 5-week-old, 7-week-old, and 12-week-old R6/2 mice and their littermate controls ($n = 6$ in each group). **c** The protein levels of Gal3 in the striatum of R6/2 and their littermate controls (12 weeks old) were analyzed by western blot analysis ($n = 4$ in each group). **d**, **e** Brain sections from WT and R6/2 mice (12 weeks old, $n = 4$ in each group) were stained with anti-Gal3 antibody (green) and a microglial marker (Ibal, red). Nuclei were stained with Hoechst (blue). Arrows mark the Ibal-positive cells (i.e., microglia). **f**, **g** Primary cultures of microglia prepared from R6/2 and littermates as described in the "Methods" section. The cells were collected 2 days after plating and were stained with anti-Gal3 antibody (red) and anti-Ibal antibody (gray). Three independent experiments were performed. **h** Representative histogram plot of WT and R6/2 microglia stained with Gal3 and assessed by flow cytometry. **i** The levels of *Lgal3* transcripts in primary microglia were quantified by RT-qPCR. Three independent experiments were performed. **j** Brain sections from WT and Hdh[150Q] mice (21 months; $n = 3$ in each group) were stained with anti-Gal3 (green) and anti-Ibal antibodies (red). Nuclei were stained with Hoechst (blue). Arrows mark the Ibal-positive cells (i.e., microglia). **k** Striatal tissues of Hdh[150Q] (15 months old; $n = 3$ in each group) were carefully removed for RNA preparation and were subjected to RT-qPCR as detailed in the "Methods" section. Data in **a** and **b** were analyzed by two-way ANOVA followed by Sidak's multiple comparisons test. *$P < 0.05$. Data in **c**–**k** are presented as the means ± SEM and were analyzed by the unpaired Student's *t*-test. *$P < 0.05$, ***$P < 0.001$, ****$P < 0.0001$. n.s., not significant. Scale bar: 10 μm. Source data is available as a Source Data File

HD microglia. Similar to treatment with Bay11, Gal3-knockdown reduced the NFκB activity, as measured by NFκB-p65 transcription factor assay (Fig. 4c), and the nuclear levels of phosphor-p65 and p65 determined by western blot analyses (Supplementary Fig. 5b). Consistent with the importance of NFκB in neuroinflammation, the levels of cytokines and NO in the medium were also normalized by sh*Lgals3*. Collectively, knockdown of Gal3 markedly reduced the levels of inflammatory mediators (including IL1β, IL6, TNFα, NO) and increased the levels of an anti-inflammatory cytokine (IL10, Fig. 4d, e).

We next evaluated the role of Gal3 in microglia using TD139, a highly specific and cell-permeable Gal3 inhibitor[33,34]. ELISA showed that treatment with TD139 inhibited the inflammation in a dose-dependent manner. TD139, at 10 μM, effectively normalized the abnormal production of inflammatory cytokines by HD microglia (Fig. 5a). It is to be noted TD139 inhibits Gal3 by direct binding and that Gal3 may exist both intracellularly and extracellularly[34]. To assess the location where Gal3 exerted its function, we treated microglia with lactose, which is not cell permeable[35]. ELISA showed that, up to 100 mM of lactose or sucrose (the osmolarity control of lactose) did not affect the release of cytokines (including IL1β, IL6, TNFα, and IL10) by HD microglia (Fig. 5b), suggesting that Gal3 might not regulate the inflammatory response through an extracellular site. We also isolated microglia directly from adult HD mice at the diseased stage (12 weeks old). Similar to microglia isolated from neonatal mice, primary microglia isolated from adult HD mice expressed more Gal3, released a higher level of IL1β, and secreted a lower level of IL10 than those of adult WT microglia. Most importantly, treatment with TD139, a Gal3 inhibitor, significantly reduced the release of IL1β and enhanced that of IL10 (Supplementary Fig. 6). The percentages of IbaI-positive microglia were 90.9 ± 3.2% and 85.9 ± 2.0%, respectively, in the WT and R6/2 groups (Supplementary Fig. 6b), demonstrating that a majority of these cells were microglia. A small number of GFAP-positive astrocytes, but not NeuN-positive neurons, were observed in our microglia preparation. Importantly, all Gal3-positive signals were observed in IbaI-positive microglia (Supplementary Fig. 6a, e, f).

**Gal3 interferes with the clearance of damaged lysosomes.** To further investigate the role of Gal3, we performed immunofluorescence staining to analyze the cellular distribution of Gal3. Interestingly, we found that a significant portion of Gal3 signals in R6/2 microglia appeared as puncta (Fig. 6a, b), a signature of intracellular vesicle rupture evoked by amyloid proteins (such as mHTT)[22]. The presence of Gal3 puncta in WT microglia were imperceptible compared to R6/2 microglia. Immunofluorescence staining showed that Gal3 puncta in R6/2 microglia were

colocalized with a lysosomal marker (lysosomal-associated membrane protein 1, LAMP1 and LAMP2; Fig. 6a, b, Supplementary Fig. 7a). No colocalization of Gal3 puncta and a mitochondrial marker (succinate dehydrogenase iron–sulfur subunit, SHDB) or an autophagosome marker (microtubule-associated protein 1A/1B-light chain 3, LC3) was observed (Supplementary Fig. 7b, c). Moreover, the levels of LAMP1 in R6/2 microglia were also higher than that in WT microglia, indicating that these damaged lysosomes in R6/2 microglia might not be cleared effectively (Fig. 6c)[36]. Consistently, transmission electron microscopy analyses revealed that, when compared with those in WT microglia, some of the lysosome-like structures in R6/2 microglia appeared larger and abnormal (Supplementary Fig. 8). Immunogold labeling using an anti-LAMP2 antibody to label lysosomes and an anti-Gal3 antibody to detect Gal3 by electron microscopy showed that Gal3 accumulated in lysosomes of R6/2 microglia (Supplementary Fig. 9b). Treating R6/2 microglia with Lenti-sh*Lgals3* (but not a control virus harboring sh*GFP*) greatly reduced the expression of Gal3 and normalized the up-regulation of LAMP1 (Fig. 6d, Supplementary Fig. 10). Therefore, the up-regulation of Gal3 might negatively regulate the clearance of damaged lysosomes in R6/2 microglia.

We next performed a Magic Red Cathepsin assay to determine lysosome leakage. The leakage of cathepsin from damaged lysosomes cleaves the Magic Red substrates and generates red fluorescence upon excitation. Confocal imaging analysis (Fig. 6e) and fluorescence microplate reader analysis (Fig. 6f) showed that R6/2 microglia had higher levels of red fluorescence signals than WT microglia. TD139 treatment significantly reduced the signals of Magic Red in R6/2 microglia, demonstrating that Gal3 plays a critical role in lysosome leakage.

**Knockdown of Gal3 reduces microglial activation in HD mice.** To assess whether up-regulation of Gal3 is an important pathogenesis factor of HD, HD mice (R6/2, 6 weeks old, $n = 9$-11 in each group) were intrastriatally injected with Lenti-sh*Lgals3* to reduce the expression of Gal3. We selected a genetic approach to reduce Gal3 because there is no pharmacologic Gal3 inhibitor (including TD139) that readily crosses the blood-brain barrier[19]. Seven weeks post-infection, brains were carefully harvested for further analyses. Immunofluorescence staining of the sh*Lgals3*-injected striatum revealed that microglial Gal3 levels were effectively down-regulated (Figs. 7a, b and 8a). Most importantly, expression of CD68, a microglial activation marker, was also markedly reduced, indicating that the abnormal activation of microglia was ameliorated in R6/2 mice by suppression of Gal3 (Fig. 7a). Consistent with our observations in primary microglia (Fig. 4b), knockdown of Gal3 in the striatum also decreased the

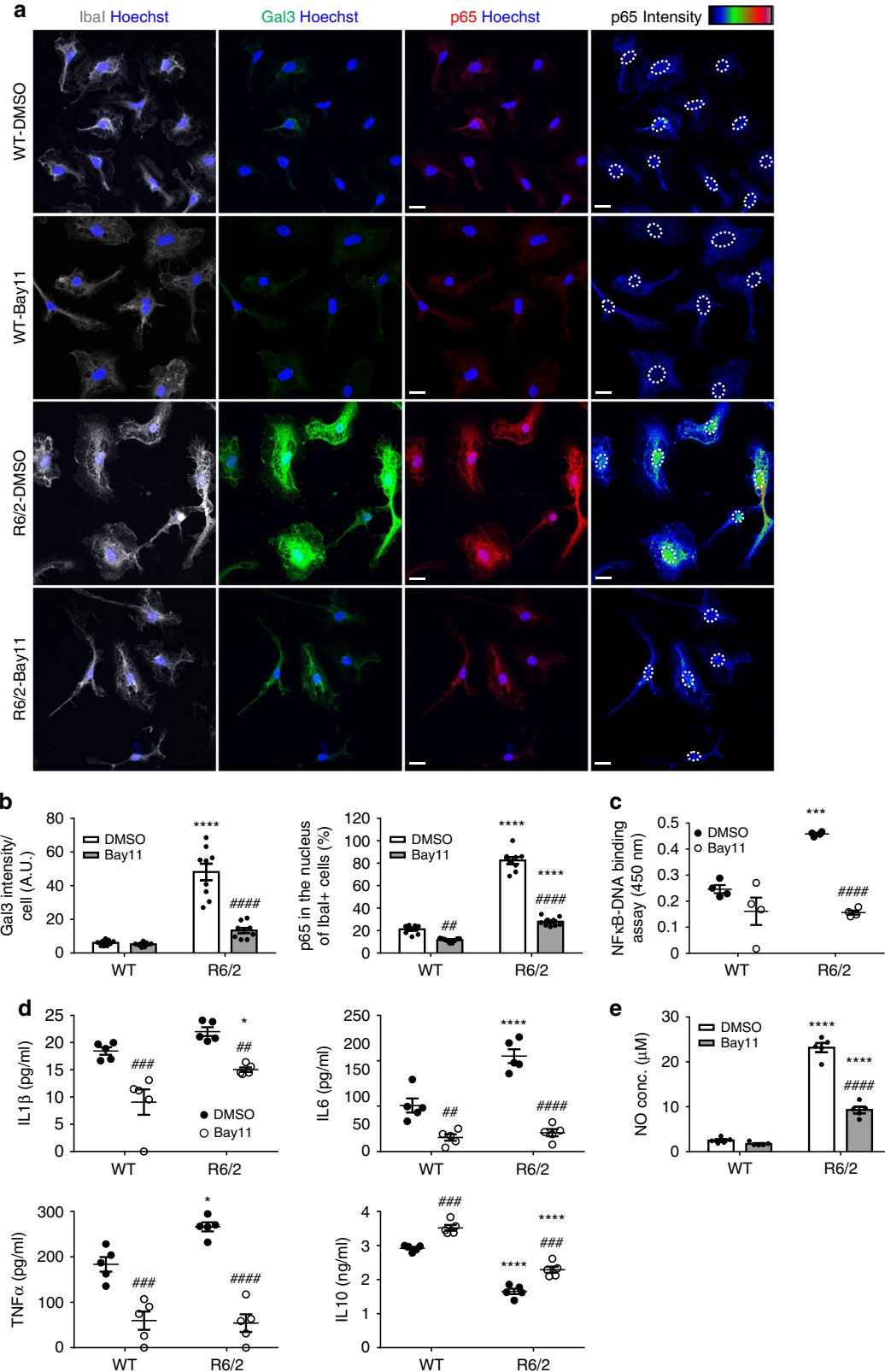

ratio of p65 in the nuclei of R6/2 microglia marked by Ibal (Fig. 7b, Supplementary Fig. 11). Most importantly, knockdown of Gal3 in the striatum of R6/2 mice reduced the level of a pro-inflammatory cytokine (IL1β) and enhanced the level of an anti-inflammatory cytokine (IL10, Fig. 7c). Elevated IL10 expression in contralateral cortical areas (i.e., the hemisphere without lentivirus

injection) was also observed (Supplementary Fig. 12). In addition, p65 was activated in Gal3-positive microglia in the striatum of Hdh$^{150Q}$ mice (Supplementary Fig. 13a).

**Gal3 triggers IL1β production via NLRP3 inflammasome.** Recent studies of macrophages have suggested that Gal3 activates

**Fig. 3** Inhibition of the NFκB pathway reduces the abnormal up-regulation of Gal3. **a, b** Primary microglia harvested from R6/2 mice and their littermate controls (WT) were cultured for 24 h and then treated with Bay11-702 (3 μM) or vehicle (0.1% DMSO) as indicated for 24 h and were then fixed for immunofluorescence staining of Gal3 (green), Iba1 (gray), and p65 (red). Nuclei were stained with Hoechst (blue). The localization of nuclei in the right-most panels is outlined by dotted lines. The color bars labeled p65 intensity represent the level of p65 intensity, from low to high fluorescence signals (blue → red, respectively). **c** An NFκB transcription factor assay was performed on the nuclear extracts prepared from the indicated primary microglia ($n = 4$). **d** ELISA was performed on the supernatants collected from the indicated primary microglia to measure the levels of IL1β, IL6, TNFα, and IL10 released by the cells. One dot represents the mean value of each sample. **e** The levels of nitrite (NO) in the supernatants were measured using the Griess reagent ($n = 5$). The results were analyzed by two-way ANOVA followed by Tukey's post hoc test. Data are presented as the means ± SEM. *Specific comparison between WT and R6/2 cells of the same treatment; #Specific comparison between the Bay11-treated and DMSO-treated groups of the same genotype; *$P < 0.05$, **$P < 0.01$, ***$P < 0.001$, ****$P < 0.0001$. Same $P$-value denotation for #. Scale bar: 10 μm. Source data is available as a Source Data File

the NOD-like receptor family, pyrin domain-containing 3 (NLRP3) inflammasome and subsequently promotes the secretion of IL1β[37]. Moreover, Gal3-null macrophages have shown impairment in the activation of the NLRP3 inflammasome and a markedly reduced production of IL1β[37]. Furthermore, lysosomal ruptures have also been reported to activate NLRP3 inflammasome[38]. We therefore assessed whether up-regulation of Gal3 in HD microglia might trigger activation of the NLRP3 inflammasome and promote the production of IL1β. Immunofluorescence assays revealed that the level of NLRP3 was up-regulated in Gal3-positive R6/2 and Hdh$^{150Q}$ microglia (Fig. 8a, Supplementary Fig. 13b). Consistent with the importance of Gal3 in the regulation of NLRP3, genetic suppression of Gal3 concurrently reduced the level of NLRP3 in vivo (Fig. 8a). In addition, treating primary microglia harvested from HD mice with an inhibitor of NLRP3 (MCC950[39]) greatly reduced the abnormal production of IL1β by HD microglia (Fig. 8b). Collectively, these data suggest that the NLRP3 inflammasome is regulated by Gal3 and mediates the production of IL1β in HD microglia.

**Depletion of Gal3 in vivo ameliorates HD symptoms**. To assess the functional outcomes of Gal3 suppression in vivo, mice were allowed to recover from the operation for 1 week before being subjected to weekly rotarod performance assays until the age of 13 weeks (end stage). As the disease progressed, HD mice injected with sh*Lgals3*-expressing virus showed significant improvements in motor performance (Fig. 9a) and lifespan (Fig. 9b, $P = 0.0089$, Kaplan–Meier survival analysis). The gradual loss of body weight was not rescued by down-regulation of Gal3 (Fig. 9c).

We have previously demonstrated that reduction of neuronal inflammation exacerbates the levels of mHTT aggregates[14]. Immunofluorescence staining showed that knockdown of Gal3 led to reduced numbers of mHTT aggregates in the striatum of HD mice (Fig. 9d, e), likely due to the normalization of cytokine levels (Fig. 7c), as previously reported[14]. No difference in the numbers of NeuN-positive cells between the striatum of WT and R6/2 mice was found (Fig. 9f), suggesting the absence of cell death in the striatum of R6/2 mice at the disease stage. This finding is consistent with previous studies, demonstrating that neuronal loss was not detected throughout the brains of R6/2 mice[30,40]. Conversely, immunofluorescence staining revealed that the presence of mHTT extensively reduces the expression of dopamine-regulated and cAMP-regulated phosphoprotein 32 kDa (DARPP32) in the striatum of R6/2 mice. Down-regulation of Gal3 markedly ameliorated the reduced levels of DARPP32 in the striatum of R6/2 mice (Fig. 9g, h). This finding is important because depletion of DARPP32 has commonly been used to indicate neurodegeneration and neuronal dysfunction in patients and mice with HD[41,42].

**Discussion**

Our findings suggest that Gal3 protein was only detected in microglia (Figs. 2c, 7a, b, and 8a), but not in astrocytes and neurons (Supplementary Fig. 2a, b, respectively), in the brains of

adult HD mice. This finding is consistent with an earlier study, indicating that Gal3 protein was only detected in microglia, but not in astrocytes and neurons of adult mouse brains[43]. It is interesting to note that Gal3 has previously been found in astrocytes harvested from neonatal mice[44]. It is possible that the expression of Gal3 might be affected by age and is expressed only in young astrocytes. Since, Gal3 up-regulation only occurred in the microglia of adult HD brains, the beneficial effects of Lenti-sh*Lgals3* are likely to result from the down-regulation of Gal3 in microglia, even though lentiviruses are capable of infecting multiple cell types in the brain. The Gal3-positive microglia studied here exhibited various morphologies, including ramified, amoeboid, and hypertrophic. Similar observations have been found in aged rhesus monkey, in which the numbers of Gal3-positive microglia with amoeboid and hypertrophic morphology are associated with cognitive impairments[45]. The expression patterns of Gal3 in Hdh$^{150Q}$ mice are consistent with the findings in R6/2 mice. Interestingly, some microglia of aged WT mice (21 months old) also expressed Gal3, but to a much lesser extent than Hdh$^{150Q}$ mice, suggesting that aging might contribute to the up-regulation of Gal3 in microglia of WT animals[45].

Using two inhibitors (Bay11 and Ro106) of the NFκB pathway, we demonstrated that NFκB plays a critical role in the up-regulation of Gal3 in HD microglia (Fig. 3a, Supplementary Fig. 4). Of note, NFκB motifs have been identified in the promoter region of Gal3[46]. Interestingly, because Gal3-null macrophages have lower NFκB responses upon stimulation[47], Gal3 is considered necessary for NFκB activation. For example, treatment with GCS-100 (a Gal3 inhibitor) reduces the levels of phosphorylated-IκB and the downstream NFκB-p65 subunit in RPMI 8226 cells[48], confirming the importance of Gal3 in NFκB activation. Hence, it is possible that NFκB and Gal3 might be regulated in a positive feed-forward loop in microglia.

Although Gal3 has been implicated in the NFκB-mediated inflammatory response, the location where Gal3 exerts its function remains elusive. Burguillos et al. demonstrated that exogenous Gal3 (1 μM) when introduced extracellularly to cells could evoke an inflammatory response in primary microglia and a microglial cell line (BV2). The authors showed that Gal3 bound to toll-like receptor 4 (TLR4) and suggested that the extracellular action of Gal3 is mediated by binding to TLR4, which triggers the subsequent inflammatory response. Treatment with an anti-Gal3 neutralizing antibody blocks the lipopolysaccharides (LPS)-induced inflammatory response, suggesting that the extracellular Gal3–TLR4 interaction might be critical for the promotion of inflammation[16]. In the present study, no exogenous Gal3 was included in the experimental conditions tested. In addition to cytokines, primary microglia also secreted Gal3 that could be detected in the conditioned medium (WT, 59.5 ± 11.0 pM; HD, 22.2 ± 2.3 pM; $n = 6$. Mean ± SEM, $P = 0.0075$, Student's $t$ test). Nonetheless, the amount of Gal3 in the conditioned microglial medium was much lower than that is needed to activate TLR4 (1 μM)[16]. Another interesting observation is that HD microglia secreted less Gal3 than WT microglia, probably owing to the

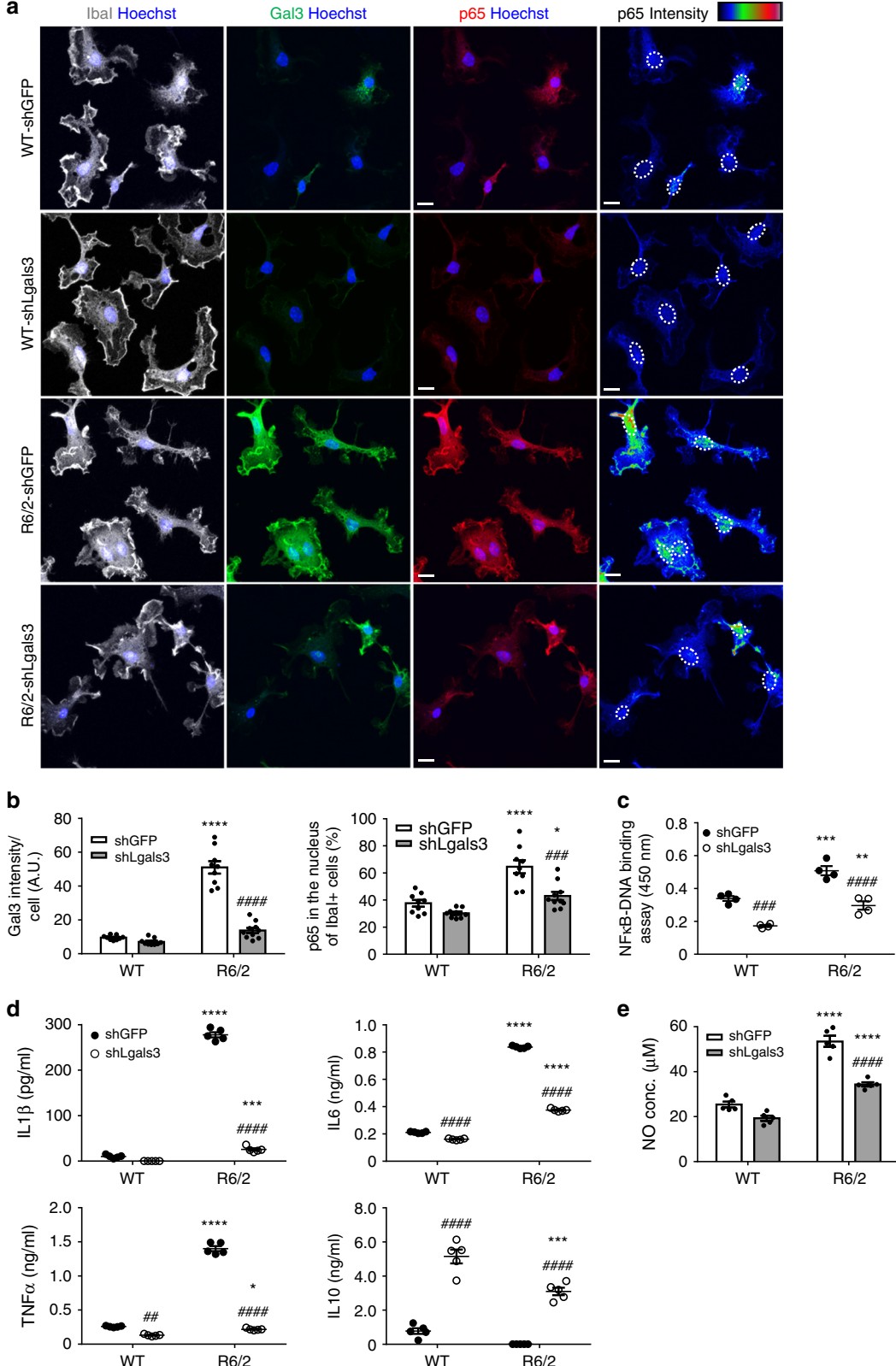

interference of mHTT on the secretion machinery[49]. Although HD microglia secreted lower amounts of Gal3 than WT microglia, they had higher inflammatory responses. The amount of extracellular Gal3 therefore seems irrelevant to the status of microglial inflammation. We also treated primary microglia with cell-impermeable lactose, which blocks extracellular Gals[35], and

found no effect on the production of cytokines in primary HD microglia (Fig. 5b). Collectively, these observations suggest that Gal3 is unlikely to exert its pro-inflammatory function through an extracellular site in HD microglia.

One emerging function of Gal3 is to monitor the integrity of endosomes and lysosomes. Upon damages of these organelles,

**Fig. 4** Suppression of Gal3 reduces the activation of NFκB and inflammation. **a**, **b** Primary microglia were infected with the indicated lentiviruses expressing shRNA against Gal3 (i.e., sh*Lgals3*) to knockdown Gal3 or the control lentivirus (sh*GFP*), as described in the "Methods" section. One week after infection, the levels of Gal3 (green) and nuclear p65 (red) were analyzed by immunofluorescence staining. Nuclei were stained with Hoechst. The localization of nuclei in the right-most panels is outlined by dotted lines. The color bars labeled p65 intensity represent the level of p65 intensity, from low to high fluorescence signals (blue → red, respectively). **c** An NFκB transcription factor assay was performed on the nuclear extracts prepared from the indicated primary microglia ($n = 4$). **d**, **e** ELISA and NO assays were performed on the supernatants collected from the indicated primary microglia to measure the levels of IL1β, IL6, TNFα, IL10, and nitrite released by the cells. One dot represents the mean value of each sample. The results were analyzed by two-way ANOVA, followed by Tukey's post hoc test. Data are presented as the means ± SEM. *Specific comparison between WT and R6/2 cells infected with the same lentivirus; #Specific comparison between the sh*GFP*-infected and sh*Lgals3*-infected groups of the same genotype; *$P < 0.05$, **$P < 0.01$, ***$P < 0.001$, ****$P < 0.0001$. Same *P*-value denotation for #. Scale bar: 10 μm. Source data is available as a Source Data File

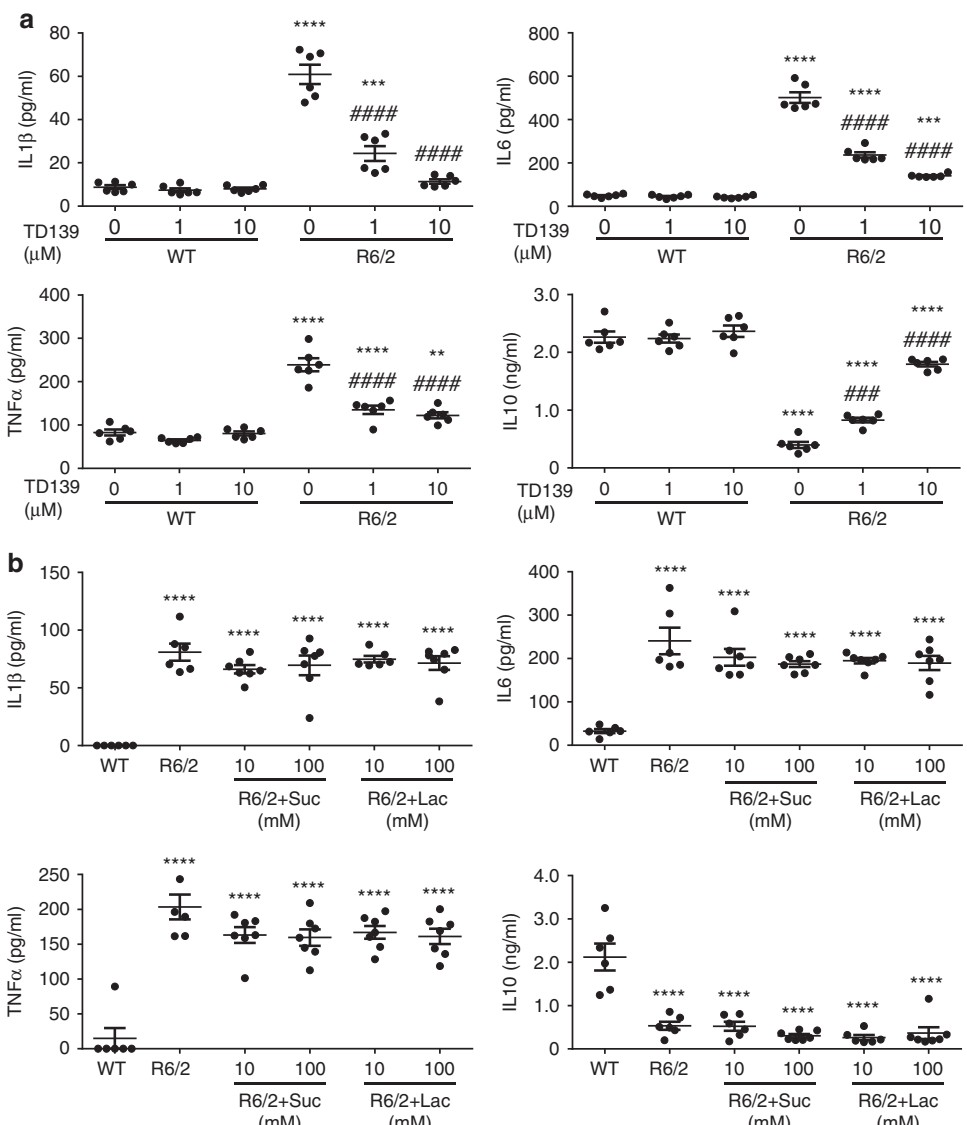

**Fig. 5** Inhibition of Gal3 by TD139 intracellularly suppresses microglial inflammation. **a** Primary microglia were cultured for 24 h and then treated with a cell-permeable Gal3 inhibitor, TD139 (1 and 10 μM), or vehicle (0.1% DMSO) for 48 h, and the supernatants were then collected for measurement of IL1β, IL6, TNFα, and IL10 levels using ELISA ($n = 6$). **b** Primary microglia were treated with lactose (10 and 100 mM) to block the binding of extracellular Gal3. Sucrose served as the osmolarity control. The supernatant was collected and subjected to ELISA ($n = 6$–7). Suc sucrose, Lac lactose. The results of **a** were analyzed by two-way ANOVA followed by Tukey's post hoc test. *Specific comparison between WT and R6/2 cells of the same treatment; #Specific comparison between the DMSO-treated and TD139-treated groups of the same genotype; $P < 0.01$. Results in **b** were analyzed by one-way ANOVA followed by Tukey's post hoc test. *Specific comparison between WT and R6/2 cells; #Specific comparison between R6/2 cells treated with Lac and Suc. Data are presented as the means ± SEM from the indicated sets of cells. **$P < 0.01$, ***$P < 0.001$, ****$P < 0.0001$. Same *P*-value denotation for #. Source data is available as a Source Data File

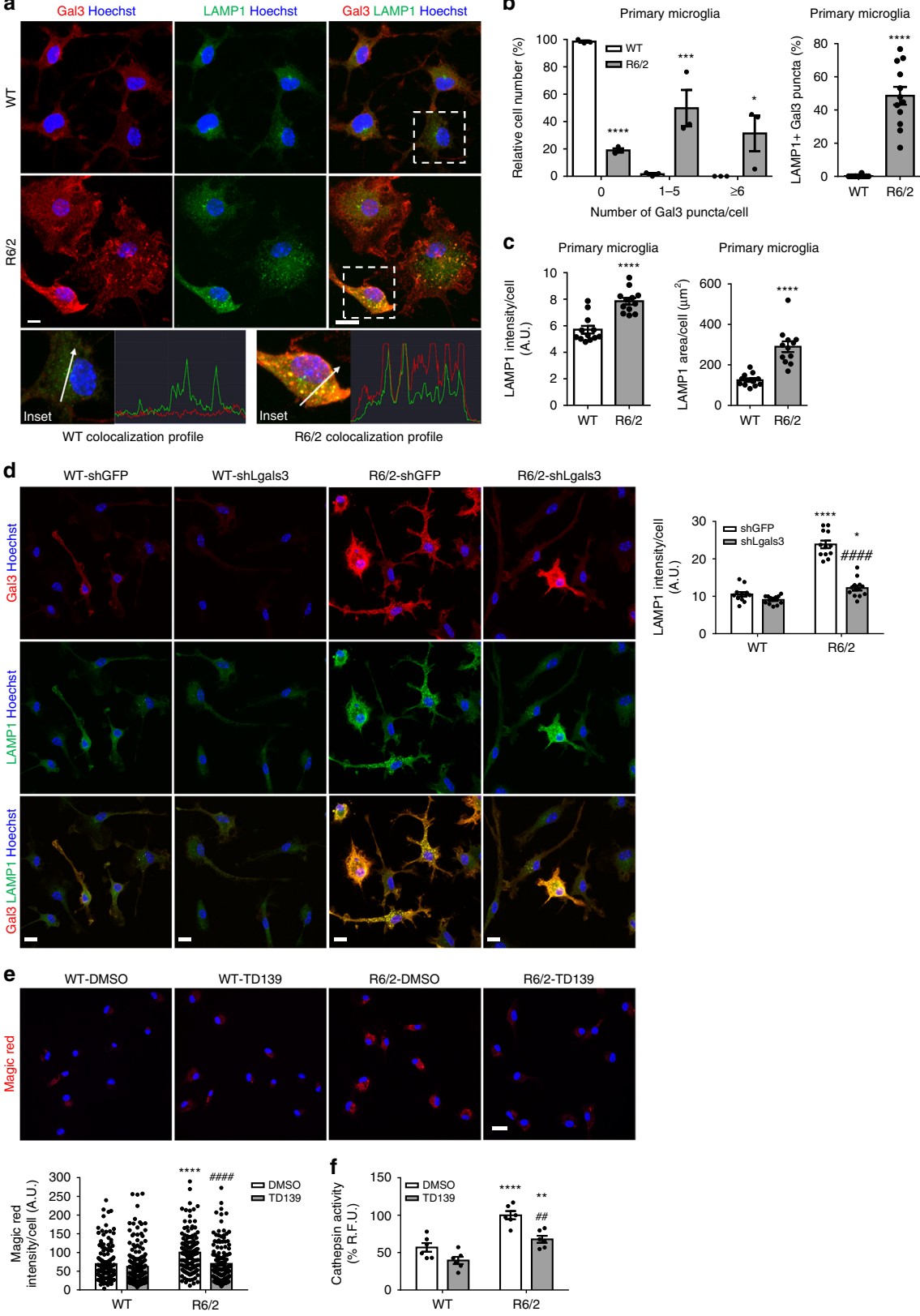

cytosolic Gal3 has been shown to redistribute and form puncta due to its binding to the glycans exposed from ruptured vesicles in several cell lines (e.g., SH-SY5Y, and N27[22,50]). The potential involvement of vesicle rupture in neuroinflammation is of great interest because a recent report suggested that lysosomal vesicle rupture increases the inflammasome activation in neuronal (SH-SY5Y) and monocytic cell lines (THP-1)[50]. Furthermore, the uptake of several disease-causing proteins (including α-synuclein, tau or mHTT) of neurodegenerative diseases evoked vesicle rupture in SH-SY5Y cells, as monitored by Gal3 puncta formation[22]. To our best knowledge, we are the first to report on the discovery of endogenous Gal3 puncta formation in microglia of

**Fig. 6** Knockdown of Gal3 improves the clearance of damaged lysosomes. **a** Primary cultures of microglia were prepared as described in the "Methods" section. The cells were harvested 2 days after plating and stained with a lysosomal marker (LAMP1, green) and anti-Gal3 antibody (red). **b** The percentages of microglial cells exhibiting Gal3 puncta (6–10 pixels in width) were quantified from more than 100 cells in each group. Galectin-3 puncta is the marker of vesicle rupture. **b, c** The signals of LAMP1 and the percentage of colocalization with Gal3 puncta were quantified. Data are presented as the means ± SEM and were analyzed by the unpaired Student's *t*-test. **d** Primary microglia were infected with the indicated lentiviruses expressing shRNA against Gal3 (i.e., sh*Lgals3*) to knockdown Gal3 or the control lentivirus (sh*GFP*), as described in the "Methods" section. One week after infection, the levels of Gal3 (red) and LAMP1 (green) were analyzed by immunofluorescence staining. Nuclei were stained with Hoechst. **e, f** Primary microglia were treated with TD139 (10 μM) or vehicle (0.1% DMSO) for 48 h. After 48 h, cells of the indicated treatment were incubated with the Magic Red solution for 30 min and visualized by confocal microscopy **e** or monitored by a fluorescence microplate reader as described in the "Methods" section **f**. The results were analyzed by two-way ANOVA, followed by Tukey's post hoc test. Data are presented as the means ± SEM. *Specific comparison between WT and R6/2 cells infected with the same lentivirus or treatment; #Specific comparison between the sh*GFP*-infected and sh*Lgals3*-infected groups or DMSO-treated and TD139-treated groups of the same genotype; *$P < 0.05$, **$P < 0.01$, ***$P < 0.001$, ****$P < 0.0001$. Same *P*-value denotation for #. Scale bar: 10 μm. Source data is available as a Source Data File

neurodegenerative disease. In the present study, we detected that Gal3 puncta were colocalized with lysosomes (LAMP1 and LAMP2) in primary HD microglia (Fig. 6a, b, Supplementary Fig. 7a) that contain mHTT intracellularly. mHTT might trigger lysosome rupture, allowing Gal3 to bind to β-galactoside-containing glycoconjugates originally located in the luminal side of lysosomes and form puncta. This interferes with the clearance of damaged lysosomes and ultimately contributes to the over-activation of the neuroinflammatory response in HD microglia.

Ruptured vesicles are generally directed towards autophagic degradation. A recent study by Yoshida et al. demonstrated that, during lysosome damage, at least five lysosomal proteins (including LAMP1 and LAMP2) are ubiquitinated, which facilitates the recruitment of autophagic machinery for the subsequent clearance[51]. In HD microglia (R6/2), we did not observe the colocalization of Gal3 puncta and LC3 (Supplementary Fig. 7b), suggesting that the autophagic machinery might not be success-fully recruited to the ruptured lysosomes to induce lysophagy[51]. Interestingly, Gal3 has been shown to bind with LAMP1[52]. It is possible that the formation of Gal3 puncta in damaged lysosomes might inhibit the formation of lysophagy by interfering with ubiquitination of lysosomal proteins in damaged lysosomes. Our data showed that down-regulation of Gal3 normalized the excess lysosomal mass, further supporting that Gal3 might negatively regulate the clearance of damaged lysosomes. The accumulation of ruptured lysosomes in HD microglia might contribute to the activation of NLRP3 inflammasome[38].

Our findings also suggest an important role for NLRP3 inflammasomes in the feedback loop between Gal3 up-regulation and NFκB in HD microglia. The role of NLRP3 in neurodegenerative diseases has not yet been extensively investigated. Heneka et al. have reported that activation of NLRP3 occurs in the brains of mice with Alzheimer's disease (AD) and mediates the maturation of IL1β and subsequent inflammatory events. NLRP3 appears to be important in AD pathogenesis because genetic removal of NLRP3 in APP/PS1 mice drastically reduces AD pathology[53]. To the best of our knowledge, the function and regulation of NLRP3 in HD has not yet been established. Previous studies have shown that Gal3 directly interacts with NLRP3 in activated macrophages and promotes the activation of NLRP3 inflammasome[54]. In the present study, we demonstrated that suppression of Gal3 concurrently attenuates the activation of NLRP3 inflammasomes in HD microglia. Noting that Gal3 suppression did not completely inhibit the amount of NLRP3 in R6/2 microglia (Fig. 8), and NFκB can activate NLRP3 inflammasomes directly[55], we cannot rule out the possibility that, in addition to the Gal3-dependent pathway, NFκB might also activate NLRP3 inflammasome directly in a Gal3-independent pathway in HD microglia.

We have previously demonstrated that reduction in the inflammatory response leads to the reduced amounts of mHTT

aggregation[14]. Consistent with this earlier finding, we showed in the present study that genetically reducing Gal3 in the striatum of HD mice not only reduced inflammation in the brains of R6/2 mice but also reduced the amounts of mHTT aggregates (Fig. 9d, e). Because inflammation has been shown to negatively regulate autophagy[56], the suppression of Gal3-mediated inflammatory response might protect neuronal cells from inflammation, normalize the activity of autophagy to degrade mHTT, and decrease mHTT aggregation. Most importantly, down-regulation of microglial Gal3 ameliorated the reduced expression of DARPP32 in HD MSN (Fig. 9g, h), suggesting the rescue of neuronal functions. This is an important finding because mHTT dose-dependently reduces the expression of DARPP32[41], and the number of DARPP32-positive neurons is negatively correlated with HD Motor Impairment Score[42].

Our study demonstrated that HD patients had higher levels of plasma Gal3 than non-HD subjects (Fig. 1a). Importantly, the elevated plasma Gal3 levels are associated with the clinical features of HD. Such an up-regulation of plasma/serum Gal3 levels has also been observed in several other brain diseases[24–26]. On the other hand, the elevated plasma Gal3 has been reported to show positive correlation with cardiac and systemic inflammation, and potentially serve as biomarker for patients with myocardial infarction. Alturfan et al. has demonstrated that Gal3 as a bio-marker associated with inflammation in acute myocardial infarction. Besler et al. on the other hand has reported myocardial Gal3 as a biomarker to reflect cardiac inflammation and fibrosis, while plasma Gal3 correlated with inflammatory cell counts[57,58]. Earlier studies also suggest that myocardial infarction is relevant to several types of neuropathology, such as dementia and migraine, which are tightly associated with neuroinflammation[59,60]. HD patients exhibit chronic systemic inflammation due to the expression of mHTT in blood cells (such as macrophages), which are more reactive to cytokine produc-tion[61]. Up-regulation of plasma Gal3 appears to be nonselective for multiple neurodegenerative diseases. Therefore, the level of plasma Gal3 by itself might not serve as a good biomarker. A combination of multiple biomarkers, such as plasma Gal3 plus other disease-specific markers (e.g., the level of plasma mHTT for HD), is needed to design reliable biomarker sets to monitor the disease progression of HD and other neurodegenerative diseases.

Similar to the findings in human plasma, plasma Gal3 levels in R6/2 mice were significantly up-regulated at the disease mani-fested stage (12 weeks old). The cellular source(s) that contribute to the elevated plasma Gal3 and the underlying mechanism for the release of Gal3 in HD remain unclear. Given that the disease-causing transgene originated from the human HD gene, which included ~1 kb of the promoter element and exon 1 of the *htt* gene[30], R6/2 mice exhibited many symptoms and pathologies that are parallel to HD patients (including up-regulation of Gal3 in the brain and in peripheral circulation). Such up-regulation of Gal3

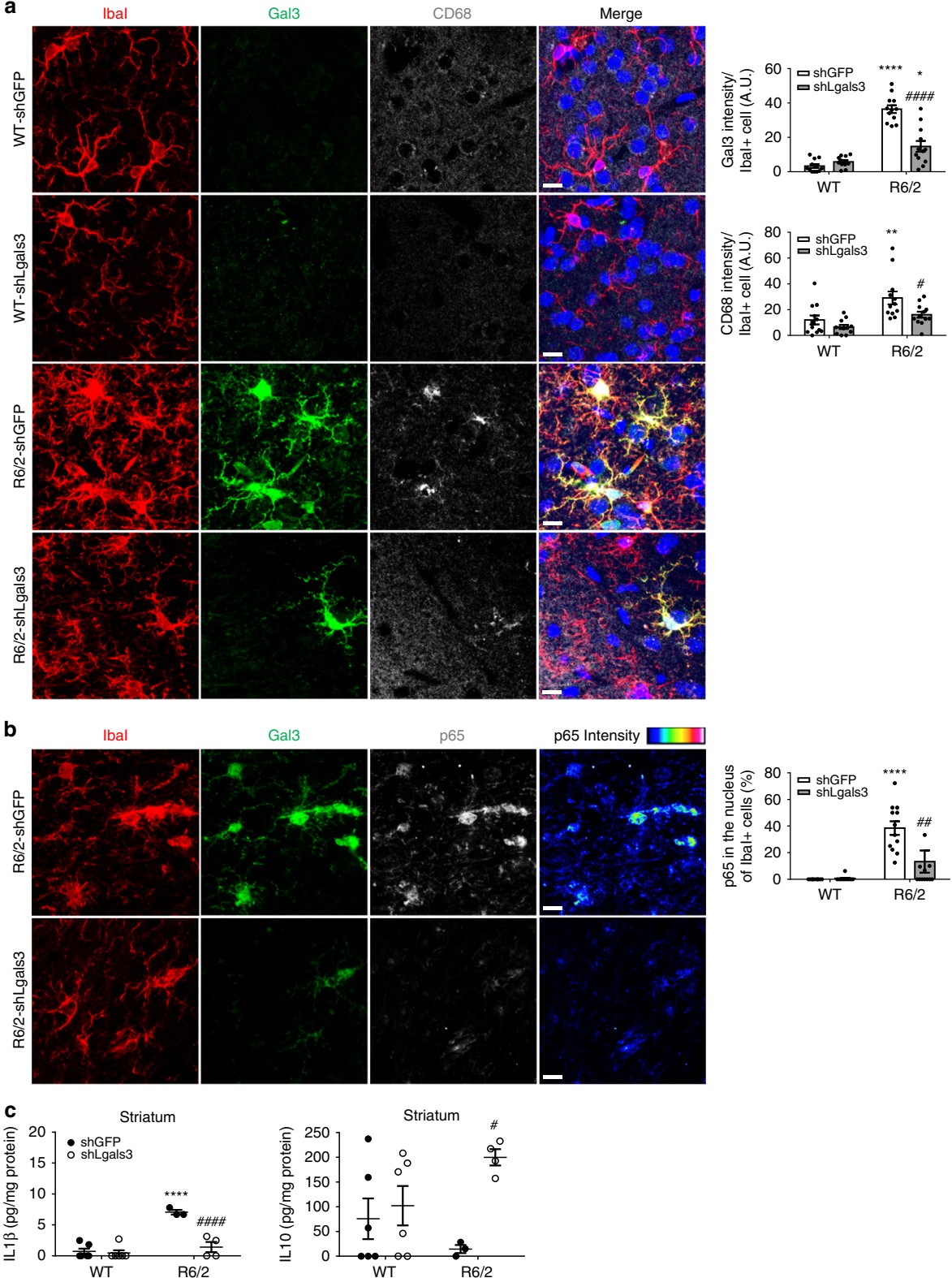

in the plasma might be due to the expression of mHTT in blood cells of humans and mice with HD[61], which has been shown to enable blood cells (particularly myeloid lineage cells, such as macrophages) to be more reactive for cytokine production in an NFκB-dependent pathway. Ample evidence suggests that various peripheral blood cells (including macrophages) are able to express Gal3 upon stimulation[62]. The enhancement of plasma Gal3 in patients and mice with HD might be attributed to the abnormally activated peripheral cells (such as macrophages) and/or microglia in the brain, which is worth further investigation. The functional impact of the suppression of Gal3 in blood cells on HD progression is of great interest and certainly warrants future investigation.

As summarized in Fig. 10, our findings reveal a previously unknown role of microglial Gal3 in HD pathogenesis and provide a new target for the development of novel therapeutic treatments.

**Fig. 7** Knockdown of Gal3 in the brains of R6/2 mice reduces the activation of microglia. Mice of 6 weeks were intrastriatally injected with lentiviruses carrying the indicated shRNA and monitored for an additional 7 weeks. Brain tissues were carefully harvested and subjected to immunofluorescence assays using the indicated antibodies. Nuclei were stained with Hoechst (blue). **a** The expression levels of Gal3 (green), Iba1 (red, a microglia marker), and CD68 (gray, a marker for activated microglia) in the striatum of the indicated mice were determined using immunofluorescence staining. **b** The expression levels of Gal3 (green), Iba1 (red), and p65 (gray) in the striatum of the indicated mice were determined using immunofluorescence staining. The color bars labeled p65 intensity represent the level of p65 intensity, from low to high fluorescence signals (blue → red, respectively). Twelve image frames of each animal were analyzed (4 animals in each group). **c** Striatal homogenates were analyzed to determine the levels of IL1β and IL10 by ELISA. Each dot represents the mean value of each mouse (3–6 animals in each group). Data are presented as the means ± SEM. The results were analyzed by two-way ANOVA followed by Tukey's post hoc test. *Specific comparison between WT and R6/2 mice infected with the same lentivirus; #Specific comparison between mice of the same genotype. *$P < 0.05$, **$P < 0.01$, ***$P < 0.001$, ****$P < 0.0001$. Same $P$-value denotation for #. Scale bar: 10 μm. Source data is available as a Source Data File

Specifically, development of blood-brain barrier-permeable Gal3 inhibitors is a promising strategy worthy of further investigation.

## Methods

**Animals**. R6/2 (B6CBA-Tg(HDexon1)62Gpb/1J) and Hdh[150Q] (B6.129P2-Htttm2Detl/150J) mice were originally purchased from the Jackson Laboratory (Bar Harbor, ME, USA) and were maintained as colonies in the animal facility of the Institute of Biomedical Science under standard conditions. Mice were fed LabDiet® and housed in standard conditions, with a 12-h light:dark cycle. Genotypes of the offspring were identified by amplification of mHTT gene sequences from genomic DNA isolated from mouse tail clippings. Primers used in the R6/2 mice genotyping included the forward primer: 5′-CCGCTCAGGTTCTGCTTTTA-3′ and the reverse primer: 5′- GGCTGAG-GAAGCTGAGGAG-3′. For Hdh[150Q] mice, primers used in the genotyping included the forward primer: 5′- CCCATTCATTGCCTTGCTG3′ and the reverse primer: 5′-GCGGCTGAGGGGGTTGA-3′. All primers used in the study are listed in Supplementary Table 4. HD is an autosomal dominant disorder that affects both males and females. Nonetheless, rates of disease progression differ slightly between male and female mice with HD. The median survival times of female R6/2 mice are slightly longer than those of male mice[63]. To avoid potential sex differences, we chose to use only female mice in all behavioral studies.

**Primary microglia from neonatal mice and treatments**. Primary microglia were isolated from the astrocytic monolayer DIV 14 and 21 by gentle agitation based on the distinct adhesive features of microglia and astrocytes and were grown in Dulbecco's modified Eagle's medium (DMEM) (Invitrogen, Grand Island, USA) supplemented with 10% heat-inactivated fetal bovine serum (FBS), 2 mM L-glutamine and 1% penicillin/streptomycin at 37 °C in a humidified 5% $CO_2$-containing atmosphere. The harvested cells were plated onto poly-L-lysine-coated dishes or coverslips, and ~99% of cells were Iba1-positive as determined by immunocytochemical staining. Flow cytometry analysis showed that 99.70 ± 0.03% and 99.50 ± 0.05% of WT and R6/2 microglia were CD11b-positive cells, respectively. For experimentation, primary microglia were seeded at $7.5 × 10^4$ cells/well in 24-well plates or $2.0 × 10^5$ cells/well in six-well plates for either transduction, treatment, or cytokine release analyses. All experimental treatments were performed 1 or 2 days after seeding. All comparative experiments between HD and WT groups have been carried out using microglia isolated and cultured simultaneously and under the same conditions. Cells were treated with BAY11-7082 (sc-200615, Santa Cruz, USA), Ro 106-9920 (1778, Tocris Bioscience, USA), and MCC950 (CP-456773, Selleckchem, USA) in respective experiments. TD139 was synthesized and characterized as detailed elsewhere[33].

**Lentivirus transduction**. Lentiviruses sh*GFP* (TRCN0000072181) and sh*Lgals3* (TRCN0000054867) were produced by the National RNAi Core Facility, Academia Sinica. Primary microglial cells were transduced with the respective lentivirus at a multiplicity of infection (MOI) of 100 in the presence of 8 μg/ml polybrene (H9268, Sigma, USA). Equal volumes of fresh medium were added after 24 h. The culture media were collected for ELISA and cells were fixed for immunochemistry one week post-transduction.

**Intrastriatal injection of lentivirus**. To suppress the expression of Gal3 in vivo, lentiviruses (sh*Lgals3* and sh*GFP*) were delivered to the striatum. Briefly, R6/2 mice and littermates 5–6 weeks of age were anesthetized by an i.p. injection with ketamine/xylazine HCl (87.56 and 7.5 mg/kg, respectively) and immobilized on rodent stereotaxic frames. A burr hole was used to perforate the skull, and lentiviruses (2 μl per injection spot, $10^6$ TU/μl) were injected into the striatum using a 10 μl Hamilton syringe (Hamilton, Reno, NV, USA) at a rate of 500 ng/min. Injections into the striatum were at coordinates anteroposterior (AP) +0.8 mm, lateral (L) −2.4 mm and dorsoventral (DV) −2.7, −3.2, −3.7 mm relative to the bregma and dura surface. All procedures were performed to minimize discomfort. The mice were allowed to recover from the operation for 1 week before further

experimentation. For all subsequent IHC analyses on the effects of Gal3 knockdown, brain sections used were within a 180 μm radius from the injection site. Three fields from each brain sections, left, right, and below the injection site, were selected for analyses. Imaging was avoided at the center of the needle track due to nonspecific background signals.

**Body weight and rotarod assessment**. The body weights of the mice were measured and recorded three times weekly from 7 to 13 weeks of age. At the same time, the motor coordination was measured by a rotarod apparatus (UGO BASILE, Comerio, Italy). In brief, mice were trialed 3 days per week and three times per day for a maximum of 120 s, each at a constant speed of 12 rpm. The longest period of sustained activity on the rotarod out of three trials represented the score of the day. The averages of the highest score from 3 days represented the score of the week.

**Enzyme-linked immunosorbent assay (ELISA)**. Brain homogenates from each section, including the lentivirus-injected hemisphere of the cortex and striatum and the non-injected hemisphere of the cortex and striatum, were prepared by tissue extraction buffer (100 mM Tris, pH 7.4, 150 mM NaCl, 1 mM EGTA, 1 mM EDTA, 1% Triton X-100, and 0.5% sodium deoxycholate with freshly added phosphatase inhibitor cocktail and Protease inhibitor cocktail). The levels of IL1β, IL6, IL10, and TNFα from brain samples and supernatants collected from primary microglia cultures were determined by using respective mouse ELISA-Ready-Set-Go!® systems (Affymetrix eBioscience, USA) following the manufacturer's instruction. For brain samples, 150 μg of lysates was plated per well. For cell supernatant samples, 100 μl of the total reaction volume was two-fold diluted for IL1β, IL6, IL10, and TNFα and six-fold diluted for Gal3 detection. Gal3 from human plasma samples and cell culture supernatants were determined via DuoSet® ELISA Development system (R&D System, USA) following the manufacturer's instructions. The optical density at 450 nm was determined by a Bio-Rad model 680 (Bio-Rad, Hercules, CA, USA) or a SpectraMax 190 (Molecular Devices, Sunnyvale, CA, USA) microplate reader, with subtraction at 570 nm as the wavelength correction.

**Nitrite quantification assay**. The nitric oxide levels in supernatants were measured by using a modified version of the Griess reagent (G4410, Sigma, USA). Samples were incubated with equal volumes of Griess reagent for 15 min at room temperature (RT). Absorbance was measured at 540 nm by using a Molecular Devices SpectraMax 190 microplate reader. Quantification of the nitric oxide levels were calculated via reference to a standard curve prepared from sodium nitrite (72586, Sigma, USA) ranging from 0 to 200 μM.

**NFκB-p65 transcription factor assay**. Nuclear extracts from primary microglia were prepared with NE-PER nuclear and cytoplasmic extraction reagents (78833, Thermo Fisher, USA). The purities of the cellular fractions were determined by western blot analysis, in which lamin B1 and tubulin serve as the markers for the nuclear and cytoplasmic fractions, respectively. For the measurement of NFκB activity, nuclear extracts (10 μg per reaction) were analyzed using the NFκB-p65 Transcription Factor Assay kit (ab133112, Abcam) according to the manufacturer's protocol.

**Magic Red cathepsin assay**. Cells were treated with 1X Magic Red solution (#6134, ImmunoChemistry Technologies, USA) for 30 min, rinsed with PBS prewarmed to 37 °C, fixed and subjected to confocal microscopy analyses. Alternatively, cells were lysed in the TRIS–EDTA buffer (10 mM Tris–HCl, 1 mM disodium EDTA, pH 8.0, 0.2% Triton X-100). The signals of Magic Red were measured by a fluorescence microplate reader (SpectraMAX GEMINI EM, Molecular Devices, USA) at 622 nm.

**Immunochemical staining and quantification**. Immunofluorescence staining was performed as described[64]. Briefly, experimental animals were anesthetized by i.p. injection of 80 mg/kg of sodium pentobarbital, followed by transcardial perfusion with saline followed by 4% paraformaldehyde. Brain samples were post-fixed in 4%

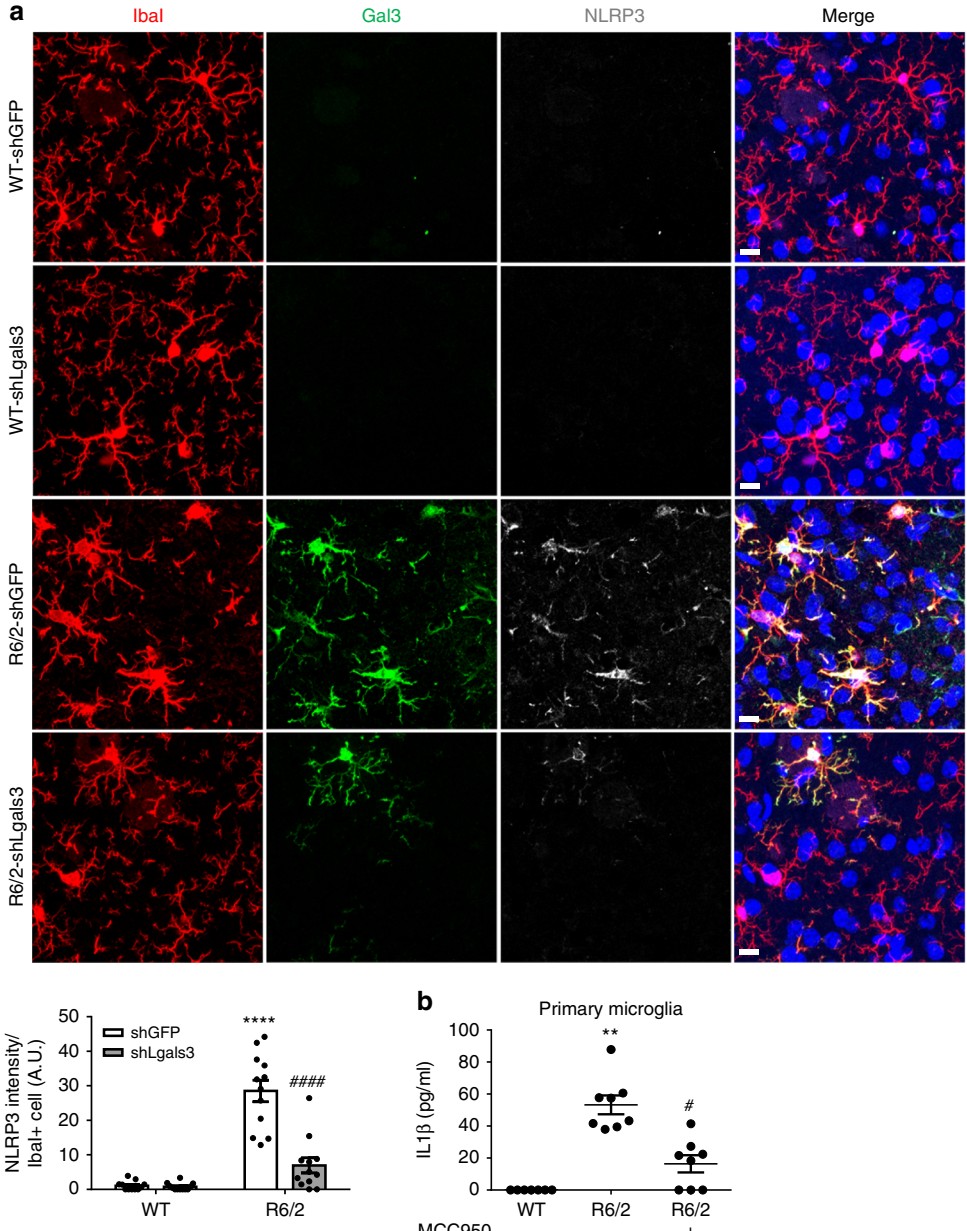

**Fig. 8** Gal3 triggers IL1β production via an NLRP3 inflammasome-dependent pathway. **a** Mice of 6 weeks were intrastriatally injected with lentiviruses harboring sh*Lgals3* to down-regulate Gal3 and then monitored for an additional 7 weeks. Brain tissues were carefully removed and subjected to immunofluorescence staining to determine the levels of Gal3 (green) and NLRP3 (gray). Four animals in each group were examined. Data are presented as the means ± SEM. *Specific comparison between WT and R6/2 mice infected with the same lentivirus; #Specific comparison between mice of the same genotype. **b** Primary microglia were treated with MCC950 (1 μM, an inhibitor of NLRP3) or PBS (0.1%) for 24 h. The levels of IL1β released by these cells were measured by ELISA. Each dot represents the mean value of one sample. Three independent experiments were conducted. Data are presented as the mean ± SEM. The results were analyzed by two-way ANOVA followed by Tukey's post hoc test. *Specific comparison between WT and R6/2 cells; #Specific comparison between R6/2 cells treated with PBS and MCC950. *$P < 0.05$, **$P < 0.01$, ****$P < 0.0001$. Same $P$-value denotation for #. Scale bar: 10 μm. Source data is available as a Source Data File

paraformaldehyde overnight at 4 °C and equilibrated in 30% (w/v) sucrose for 2 days. The tissues were transected coronally at 20 μm widths with a sliding microtome and stored in 0.1 M phosphate buffer (Na-PB, pH 7.4) with 0.1% (w/v) sodium azide. Primary cell samples were fixed in 4% (w/v) paraformaldehyde and 4% (w/v) sucrose in PBS. Brain sections and cells were permeabilized by 0.2% (v/v) Triton X-100 and 0.05% (v/v) NP-40, respectively, and blocked in 4% (w/v) BSA for 1 h. After extensive washing, samples were incubated with primary antibody at 4 °C in a humidified chamber for 1–2 days, followed by 2 h of incubation with the corresponding secondary antibody at RT. Nuclei were stained with Hoechst 33258. Primary antibodies included a marker of microglia, IbaI (1:500 for mouse and human brain sections, 1:1000 for cell samples, 019-19741, Wako Laboratory Chemicals), an antibody for brain tissue staining of Gal3 (1:300 for brain sections,

AF1197, R&D System), antibodies for the cell staining of Gal3 (0.5 μg/ml, mouse anti-Gal3 B2C10; 1.0 μg/ml goat anti-Gal3[65,66]), and antibodies for NFκB-p65 (1:100 for tissue sections, 1:200 for cell samples, MAB3026, Millipore), LAMP1 and LAMP2 (1:500, ab24170 and 1:200, ab25339, Abcam), LC3 (1:2000, GTX127375 GeneTex), SDHB (1:500, GTX113833 GeneTex), EM48 (1:500, MAB5374, Millipore), NeuN (1:1000, ABN78 and 1:1000, MAB377, Millipore), S100 (1:1000, Z0311, Dako), CD68 (1:100, ab31630, Abcam), NLRP3 (1:200, AG-20B-0014, AdipoGen), and DARPP32 (1:50, #2302, Cell Signaling). The slides were then analyzed by confocal microscopy (LSM700 stage and LSM780 microscope and Zen 2012 software; Carl Zeiss, Germany). Unless stated otherwise, three frames from one striatal section were analyzed. At least 50 IbaI-positive microglia or 200 neurons (DARPP32-positive or NeuN-positive) as indicated from each animal

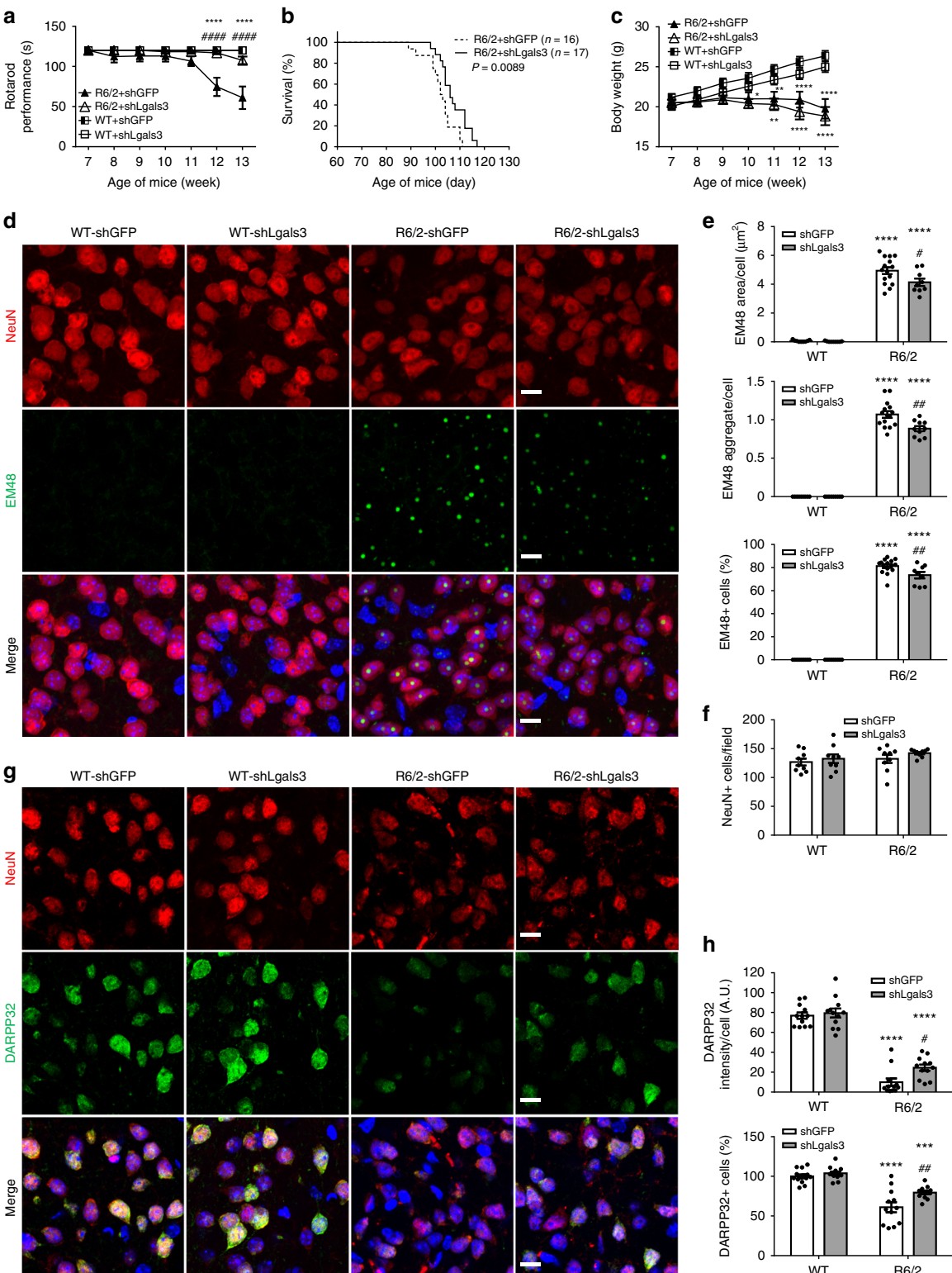

**Fig. 9** Knockdown of Gal3 ameliorates HD symptoms in R6/2 mice. Mice of 6 weeks were intrastriatally injected with the sh*Lgals3*-expressing or *shGFP*-expressing lentivirus, and monitored for an additional 7 weeks (*n* = 9–11 mice in each group). **a** The motor functions of R6/2 mice and their littermate controls were assessed by rotarod performance. **b** Kaplan–Meier plots of R6/2 and WT mice (*P* = 0.0089). **c** Body weight of the indicated animal was measured from 7 to 13 weeks of age. Brain tissues were carefully removed and subjected to immunofluorescence staining to determine the amount of mHTT aggregates (green, EM48; **d**, **e**) or DARPP32 (green, a marker for the medium spiny neurons (MSNs), **g**, **h**), on the neurons (NeuN, red; a neuronal marker; **d**, **f**, **g**). Nuclei were stained with Hoechst (blue). Four animals were analyzed in each group. The results were analyzed by two-way ANOVA followed by Tukey's post hoc test. Data are presented as the means ± SEM. *Specific comparison between WT and R6/2 mice infected with the same lentivirus; #Specific comparison between mice of the same genotype. *P* < 0.05, **P* < 0.01, ***P* < 0.001, ****P* < 0.0001. Same *P*-value denotation for #. Scale bar: 10 μm. Source data is available as a Source Data File

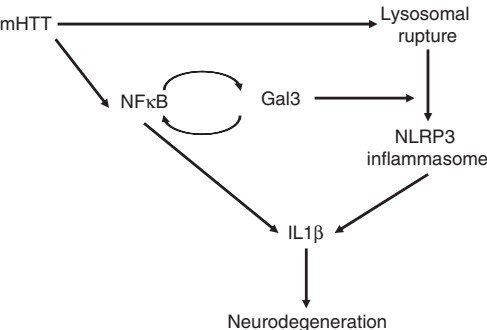

**Fig. 10** Schematic representation of the detrimental role of Gal3 in HD microglia. Under normal conditions, microglia express low levels of Gal3 and pro-inflammatory cytokines. In Huntington's disease, the presence of mHTT activates NFκB, which evokes the expression of Gal3, NLRP3, and pro-inflammatory cytokines (such as IL1β). Gal3 further activates NFκB through a positive feedback loop. On the other hand, the presence of mHTT triggers lysosomal damages as demonstrated by aggregation of Gal3 on the ruptured lysosomes. Furthermore, Gal3 inhibits the clearance of damaged lysosomes and promotes the assembly of NLRP3 inflammasomes, which results in the maturation of IL1β and subsequent inflammatory events

were scored. For quantitation, 7 or 10 z-stacks were taken for each image, with thicknesses of 10–12 μm. The settings (z-stacks and thickness) were identical when taking images for the same study. The images were analyzed by Metamorph Microscopy Automation & Image Analysis software (Molecular Devices, USA), using the multiwavelength cell scoring application for the quantitation of the average intensity per cell, colocalization application for the measurement of colocalization, and the granularity module for determining the number of puncta per cell. Detailed information on the quantification procedures are included in the Supplementary Software 1.

**RNA isolation and quantitative real-time PCR**. Total RNA was extracted by using TRIzol reagent (Invitrogen) or GENEzol™ TriRNA Pure Kit (Geneaid Biotech, Taiwan) according to the manufacturer's protocol. The isolated RNAs were transcribed to cDNA by using Superscript III (Invitrogen, USA). Quantitative real-time PCR (RT-qPCR) was performed by using SYBR® Green PCR Master Mix (Life Technologies, USA) and detected by an ABI PRISM® 7900HT Sequence Detection System (Applied Biosystems, USA). *Gapdh* and *18s* were used as housekeeping genes in mouse and human samples, respectively, to control for quantity and quality of cDNA preparations. Primers used to detect Gal3 in mouse and human samples were as follows: mouse *Lgals3*, forward primer: 5′-TTGAAGCTGACCACTTCAAGGTT-3′, reverse primer: 5′- AGGTTCTT-CATCCGATGGTTGT-3′; human *LGALS3*, forward primer: 5′-CAGAATTGCTTTAGATTTCCAA-3′, reverse primer: 5′-TTATCCAGCTTTG-TATTGCAA-3′. The gene expression levels were normalized to the respective reference gene and calculated as relative quantification. All primers used in the study are listed in Supplementary Table 4.

**Western blotting**. SDS–polyacrylamide gel electrophoresis (SDS–PAGE) and western blot analyses were performed as described[67]. Briefly, brain tissues were lysed with RIPA buffer (50 mM Tris–HCl, 0.25% sodium deoxycholate, 1% Triton X-100, 150 mM NaCl, 1× protease inhibitor cocktail and 1x phosphatase inhibitor cocktail) on ice. Lysates were cleared by centrifugation at 16,000×g for 15 min at 4 °C. Protein concentration of the collected supernatants were determined by using the Bio-Rad protein assay Dye Reagent Concentrate following the manufacturer's protocol. Samples were subjected to SDS–PAGE and transferred electrophoretically to a PVDF membrane. The membrane was blocked with 5% BSA in PBS containing 0.1% Tween-20 (v/v) (PBST), followed by incubation with the following primary antibodies (including anti-Gal3 (0.5 μg/ml, B2C10[65]), anti-lamin B1 (1:10,000, GTX103292, GeneTex), anti-tubulin (1:10,000, CP06, Calbiochem), anti-actin (1:10,000, A2066, Sigma)) overnight at 4 °C. After three washes with PBST, the membranes were incubated with the corresponding secondary antibodies for 1 h at RT. The immunoblots were visualized by using the enhanced chemiluminescence method. Uncropped data for immunoblots are provided (Supplementary Fig. 14).

**Flow cytometry**. Cells were fixed with BD Cytofix/Cytoperm solution (554722, BD Biosciences) for 20 min at 4 °C. After that, samples were washed and permeabilized twice with BD Perm/Wash buffer (554723, BD Biosciences) before sequential

labeling with a rabbit anti-Gal3 antibody (1 μg/ml)[66], a goat anti-rabbit secondary antibody (1:100, Alexa Fluor 568, A-11011, Invitrogen), and a FITC-conjugated CD11b (1:200, 101205, BioLegend) at 4 °C in the dark for 30 min. The samples were washed three times after each labeling and were resuspended in staining buffer before being subjected to an Attune® NxT acoustic focusing flow cytometer (Life Technologies, USA). Gating strategy used in the study is provided in Supplementary Fig. 15.

**Human samples**. Plasma samples, frozen at −80 °C until analysis, were obtained from 16 non-HD, 4 pre-symptomatic HD, and 26 HD individuals, recruited at Chang Gung Memorial Hospital and the Taipei Veterans General Hospital through collaboration with Dr. Chung-Mei Chen, Dr. Yih-Ru Wu, and Dr. Bing-Wen Soong, respectively. All examinations were performed in accordance with standardized protocols by a neurologist. Disease burden scores were calculated according to the formula $((CAG_n - 35.5) \times Age)$, as described[68]. The cognitive functions of HD patients were assessed by the standard MMSE[69]. The neurological function assessments of HD patients were performed according to the UHDRS[70]. Supplementary Table 1 summarizes the clinical characteristics of HD patients. Plasma samples were collected by a protocol approved by the Institutional Review Boards at Academia Sinica, CGMH, and TVGH. Written informed consent was obtained before any study-related procedures.

Human brain tissue specimens were obtained from the NIH NeuroBioBank (USA). Supplementary Tables 2 and 3 summarize the specimen demographic data for RNA preparation and immunofluorescence staining, respectively. Detailed information on human brain staining procedures are included in the Supplementary Methods.

**Statistics**. Student's t-test, standard deviation, one-way ANOVA, and two-way ANOVA were performed with the PRISM 6 program (GraphPad Software, San Diego, CA, USA). P-values < 0.05 were considered to be statistically significant. Comparisons between control and HD samples (R6/2 mice, Hdh mice, and HD patients) within the same category were made by using unpaired t-tests. Comparisons between R6/2 mice of different ages (genotype × age), primary cell treatments, animal rotarod performance, body weight changes, and ELISA results were analyzed by ANOVA (genotype × treatment) followed by Tukey's post hoc test. Data are presented as the means ± SEM. A minimum of three biological replicates was analyzed for each experimental condition. No statistical procedures were applied to pre-determine the sample sizes, but our numbers were similar to those generally practiced in the field. Correlation analyses were performed by Pearson correlation analysis to investigate the association of the plasma Gal3 levels with the clinical characteristics of HD. Kaplan–Meier analysis with a log-rank (Mantel–Cox) test was used to access mouse survival.

**Study approval**. All animal procedures were performed in accordance with the protocols of the Academia Sinica Institutional Animal Care and Utilization Committee, Taiwan. Patients were recruited in the cohort study with approval from the Institutional Review Board and the Ethics Committee of Chang Gung Memorial Hospital, Taipei Veterans General Hospital and Academia Sinica, Taiwan. Written informed consent was obtained from the participants or their next of kin.

**Reporting summary**. Further information on research design is available in the Nature Research Reporting Summary linked to this article.

## Data availability
All relevant data that support this study are available to any interested researches upon request to the corresponding author and are provided as a separate Source Data file.

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

## Acknowledgements

This work was supported by Academia Sinica and Ministry of Science and Technology, Taiwan (MOST 103-2321-B-001068, MOST 106-0210-01-15-02, MOST 107-0210-01-19-01). We thank Drs. Takashi Angata, Yi-Hsuan Lee, Jr-Wen Shui, and Guang-Chao Chen for conceptual and technical advices. We thank Mr. Yao-Kwan Huang and the Electron Microscope Laboratory of Institute of Cellular and Organismic Biology, Academia Sinica for the TEM service. We are grateful to the patients and their families, as well as normal subjects, for their participation. We thank NIH NeuroBioBank for providing the tissue samples of patients and normal subjects.

## Author contributions

J.J.S. designed, conceived, performed experiments and wrote the manuscript. H.-M.C. bred and maintained animals used in the experiments. C.-M.C., B.-W.S. and Y.-R.W. provided human plasma samples and clinical evaluations of HD patients. C.-P.C. provided technical advice and assisted human plasma sample analysis. H.-Y.C., H.-L.C. and F.-T.L. provided conceptual advice and materials. Y.-C.C. and C.-H.L. synthesized TD139. Y.C. supervised the study and edited the manuscript.
