## [Peer Review File · Nature Communications]

Reviewers' comments:

Reviewer #1 (Remarks to the Author):

The manuscript by Siew et al., entitled "Galectin-3 is required for the cell-autonomous microglial inflammation in Huntington's disease", provides evidence that upregulation of Gal3 in microglia contributed to HD pathogenesis in a feed-forward NF κ B and NLRP-3 inflammasome-dependent manner. Overall, this manuscript is high quality and important. The question is timely and therapeutically relevant. While microglial Gal3 has previously been implicated in other neurodegenerative diseases, the authors provide new mechanistic insight into the role of microglial Gal3 in HD. The potential of Gal3 as a biomarker for disease (Fig 1) is particularly powerful. Some concerns are outlined below:

1. The authors use lentivirus infection to knockdown Gal3. Because this virus seems to lack a reporter, one is not able to assess cell-type specificity or measure transduction efficiency. Given that other cell types have been reported to express Gal3 in vivo, such as astrocytes (Zhang et al. Journal of Neuroscience. 2014), the authors should better address whether the in vivo effects (Fig 9) are microglia-specific.
2. It is unclear why the authors describe lysosomes as damaged in Figure 6. This requires further characterization. For example, the authors could assess this the ultrastructure of the lysosomes by electron microscopy.
3. The authors state that, "HD mice injected with shLgals3-expressing virus showed a significant delay in motor performance deterioration". However, the data presented in Figure 9A does not reflect a delay since they never reach the motor deficits of HD mice treated with the control virus.
4. The authors should provide survival curves of the mice in Fig 9. Do shLgal3 treated HD mice have prolonged survival?
5. The EM48 and DARPP32 rescue in Figure 9 requires further quantification. The standard in the HD field is to quantify the number of EM48 aggregates per cell and the % of cells with EM48 aggregates. Additionally, the authors should quantify the % of DARPP32 positive cells in each of their experimental conditions.
6. Measures of cell death and axon degeneration would be informative in Fig 9.
7. The authors describe the mechanism in which NF-kappaB activation and NLRP3 inflammasome work through Gal3. Given NF-kappaB can activate NLRP3 inflammasome directly, this should be considered.
8. The authors show that LGAL3 mRNA is significantly increased in the caudate nucleus of HD patients (Fig 1). To better correlate these human data with their findings in mice, the authors should perform in situ hybridization or immunofluorescence for Gal3 in human tissue to show increased Gal3 in microglia.

Minor Concerns:

1. The authors show that reducing Gal3 in vivo in the R6/2 model leads to reduction of CD68 in microglia in Fig 7, however it is unclear why there is no apparent basal CD68 staining in the representative images of control conditions (WT-shGFP and WT-shLgals3).
2. The imaging methods should include more detail (how were fields chosen, were z-stacks taken and, if so, how thick, etc.). The description of the analysis of images also requires more detail.
3. The authors state that they performed T-tests in Figure 1A and Figure 2A when ANOVAs should have been performed.
4. In the introduction, the authors state, "Although upregulation of Gal3 has previously been reported in microglia, the role of Gal3 in neuroinflammation is largely uncharacterized". However, the authors then discuss 3 studies describing the role of Gal3 in neuroinflammation. This sentence should be edited.
5. In Fig 2, the authors should quantify the percentage of Gal3 positive cells that are IBA-1 positive in R6/2 mice.
6. The discussion is quite lengthy and reads more like a review article. Although thorough, this should be reduced.

Reviewer #2 (Remarks to the Author):

The manuscript describes the pathophysiological effects of galectin-3 (Gal-3) in Huntington's disease (HD) in humans, in two mouse HD models and in microglial cells from wild-type and HD mouse lines. In

humans, the authors provide evidence of significantly higher Gal-3 levels of in plasma that correlate with HD burden. In a HD mouse model, R6/2 mice, using an array of genetic-based tools (shGals3) and pharmacological agents that inhibit either Gal-3 (TD139) or potential downstream pathways (NfκB inhibitors BAY11-702 and Ro 106-9920), the authors report that microglial cells are the cellular source of Gal-3 upregulation and toxicity and that attenuated/aborted Gal-3 upregulation limits accumulation of inflammatory mediators and affects NfκB signaling. They also report neuroprotection and attenuated motor dysfunction in mice with depleted Gal-3. The manuscript provides novel information on the injurious role of Gal-3 in HD, adding to the growing number of publications attesting to the injurious role of Gal-3 in various neurodegenerative diseases. Parallel data in humans and mice is the strength of the study. The study is logically planned and the manuscript is well written.

The two key weaknesses include: 1) the lack of mechanistic link between human and animal data. Increased Gal-3 in peripheral circulation is proposed as the pathophysiological mechanism in HD in humans based on the established relationship between major increase in Gal-3 levels in plasma of HD patients and disease worsening, but no studies of the role of peripheral Gal-3 were performed in mice. Thus, the link between mouse and human is missing/undeveloped; and 2) multiple concerns related to insufficient characterization of microglial population and imprecision of methods used in studying isolated microglial cells.

The following are several major weaknesses:

- 1). Among several human subjects with pathologies unrelated to HD (Suppl Table 2), three had either myocardial infarction, seizure, possible stroke, or congestive heart failure. The latter clinical conditions are known to upregulate Gal-3 expression and contribute to respective diseases.
- 2). The caudate is believed to be the susceptible region. Fig. 1G shows ~2-fold increase in LGALS3 mRNA, whereas Suppl. Fig. 1 shows 3-6 fold increase in other brain regions in human brains. This finding should be discussed. Data in Suppl. Fig. 1 should be moved to Fig. 1.
- 3). Vastly different morphology of microglia obtained from astrocyte-microglial cultures of WT mice shown in Fig. 2D, 3A and Suppl Fig. 3. While cells are spindle-looking in Fig. 3A, they have "activated" morphological phenotype in Fig. 2D and Suppl Fig. 3. It is unclear which panels are representative. Given the presence of ruffling in some panels, it needs to be made clear whether all experiments were conducted at a single time point. Comparative study between microglial cells from WT and R6/2 mice should be performed at least at 2 time points to rule out a possibility that it takes longer for WT microglial to acquire similar morphologic phenotype as R6/2.
- 4). Proliferation rate is a plausible explanation for much higher cell density shown for microglia obtained from R6/2 mice. Gal-3 is known to affect microglial proliferation. Thus, the presence/lack of increased proliferation is critical for data interpretation in several experiments.
- 5). NfκB activity is based on immunofluorescence-based appearance of p65 in the nucleus (Fig. 3), not on actual measurements of activity. Cell fractionation assay is the preferred way for evaluating protein translocation, with appropriate demonstration of cellular fraction purity. EMSA should be used to evaluate NfκB activity.
- 6). Insufficient information on the timing for experiments in isolated microglial cells. Microglial assays are said to be run 24-48 hours after plating. The precise timing should be provided to account for possible effects of evolving cell adherence over time.
- 7) Isolation of microglial cells directly from diseased/WT, rather than isolation from neonatal astrocyte-microglial cultures, would provide more relevant information on Gal-3 mediated effects in HD.

Reviewer #1

1. The authors use lentivirus infection to knockdown Gal3. Because this virus seems to lack a reporter, one is not able to assess cell-type specificity or measure transduction efficiency. Given that other cell types have been reported to express Gal3 *in vivo*, such as astrocytes (Zhang et al. *Journal of Neuroscience*. 2014), the authors should better address whether the *in vivo* effects (Fig 9) are microglia-specific.

Response: We agree with the reviewer that lentivirus would be able to infect multiple cell types in the brain. In the reference mentioned above, astrocytes harvested from mice at the age of postnatal day 7 have been found to express *Lgals3* transcripts (Zhang et al. *Journal of Neuroscience* 34: 11929-11947, 2014), which is similar to a previous finding in primary astrocytes (Pasquini et al., *Cell Death Differ* 18: 1746-1756, 2011). Nonetheless, in the adult brains of our HD mice (R6/2) and their littermate controls at the age of 12 weeks old, Gal3 protein was only detected in microglia (Fig. 2C, 7A, B, 8A), but not in astrocytes and neurons (Supplementary Fig. 2A, B, respectively). Our finding is consistent with an earlier study indicating that Gal3 protein was only detected in microglia, but not in astrocytes and neurons of adult

mouse brains (Mok et al., *Biochem Biophys Res Commun* 359: 672-678, 2007). Thus, expression of Gal3 in astrocytes might be affected by age, and it is expressed in relatively young astrocytes. Since only microglia in adult brains express Gal3, the beneficial effects of Lenti-shLgals3 are likely to result from the down-regulation of Gal3 in microglia of adult HD brains. We have included this discussion in the revised manuscript (page 14, lines 14-23), and would like to thank the reviewer for raising this issue.

2. It is unclear why the authors describe lysosomes as damaged in Figure 6. This requires further characterization. For example, the authors could assess the ultrastructure of the lysosomes by electron microscopy.

Response: We considered that the lysosomes of HD microglia were damaged because of the appearance of Gal3 puncta in lysosomes (Fig. 6), a sensitive assay for the detection of damaged lysosomes, as described previously (Aits et al., *Autophagy* 11: 1408-1424, 2015; Freeman et al., *PLoS One* 8: e62143, 2013). We have included details in the Discussion section (page 17, lines 5-21).

To further characterize lysosomal damage, we have employed transmission electron microscopy to analyze HD microglia and found that the lysosome-like structures in HD microglia were abnormal compared to those of controls (Supplementary Fig. 8) (Lei et al., *Mol Vis* 18: 103-113, 2012; Liaury et al., *J Neuroinflammation* 9: 56, 2012).

In addition, we have performed a Magic Red Cathepsin assay and described the results in the revised manuscript (Fig. 6E, F). Leakage of cathepsin, a lysosomal enzyme, from damaged lysosomes cleaves the loaded Magic Red dye to generate red fluorescence signals. As detailed in the revised manuscript (page 11, lines 4-9), the results of the Magic Red Cathepsin assay confirmed lysosome leakage in HD microglia. We thank the reviewer for this suggestion.

3. The authors state that, “HD mice injected with shLgals3-expressing virus showed a significant delay in motor performance deterioration”. However, the data presented in Figure 9A does not reflect a delay since they never reach the motor deficits of HD mice treated with the control virus.

Response: We have modified the indicated sentence in the revised manuscript to read “...HD mice injected with shLgals3-expressing viruses showed a significant improvement in motor performance” (page 12, lines 22-23).

4. The authors should provide survival curves of the mice in Fig 9. Do shLgal3 treated HD mice have prolonged survival?

Response: As recommended, we performed a new experiment to address this issue. As shown in Fig. 9B, down-regulation of Gal3 improved the survival of R6/2 mice. We would like to thank the reviewer for this suggestion.

5. The EM48 and DARPP32 rescue in Figure 9 requires further quantification. The standard in the HD field is to quantify the number of EM48 aggregates per cell and the % of cells with EM48 aggregates. Additionally, the authors should quantify the % of DARPP32 positive cells in each of their experimental conditions.

Response: We have quantified the number of EM48 aggregates per cell, the % of cells with EM48 aggregates, and the % of DARPP32-positive cells, as recommended. The data are presented in Fig. 9E and 9H in the revised manuscript.

6. Measures of cell death and axon degeneration would be informative in Fig. 9.

Response: We have quantified the number of NeuN-positive neurons in the striatum and found no difference between WT and R6/2 mice, suggesting no obvious cell death at the end stage in R6/2 mice. The presence of mHTT did reduce the expression of DARPP32, which can be restored by down-regulation of Gal3. We have incorporated these data in the revised manuscript (pages 13, Lines 6-13; Fig. 9F, G, H).

Although axon degeneration and axon swelling have previously been reported in Hdh mice (a knock-in mouse model of HD), no axonal abnormalities were found in R6/2 mice (Li et al., J Neurosci 21: 8473-8481, 2001; Marangoni et al., Neurobiol Aging 35: 2382-2393, 2014). Therefore, we did not assess axon degeneration in the revised manuscript.

7. The authors describe the mechanism in which NF-kappaB activation and NLRP3 inflammasome work through Gal3. Given NF-kappaB can activate NLRP3 inflammasome directly, this should be considered.

Response: We agree with the reviewer. Because down-regulation of Gal3 did not completely suppress the amount of NLRP3 in R6/2 microglia (Fig. 8A), we cannot exclude the possibility that NF-kappaB may activate NLRP3 inflammasomes directly. This discussion was included in the revised manuscript (page 19, lines 6-10).

8. The authors show that LGAL3 mRNA is significantly increased in the caudate nucleus of HD patients (Fig 1). To better correlate these human data with their findings in mice, the authors should perform in situ hybridization or immunofluorescence for Gal3 in human tissue to show increased Gal3 in microglia.

Response: As recommended, we have performed immunofluorescence staining to assess the amount of Gal3 in the brains of HD patients and non-HD subjects. Our data showed that the level of Gal3 was higher in the microglia of HD patients when compared with the non-HD subjects tested (Supplementary Fig. 1).

Minor Concerns:

1. The authors show that reducing Gal3 in vivo in the R6/2 model leads to reduction of CD68 in microglia in Fig 7, however it is unclear why there is no apparent basal CD68 staining in the representative images of control conditions (WT-shGFP and WT-shLgals3).

Response: In contrast to macrophages that constantly express CD68, the level of CD68 in microglia under normal physiological conditions is very low or undetectable (Perego et al., Neuroinflammation 8: 174, 2011). Thus, there is no apparent basal CD68 staining in microglia (Fig. 7).

2. The imaging methods should include more detail (how were fields chosen, were z-stacks taken and, if so, how thick, etc.). The description of the analysis of images also requires more detail.

Response: We have added the requested information in the Methods section under the subtitle of “Immunochemical staining and quantification” (page 26, lines 20-23, page 27, lines 1-5) and “Intrastriatal injection of lentivirus” (page 23, line 22 page 24, lines 1-2).

3. The authors stat that they performed T-tests in Figure 1A and Figure 2A when ANOVAs should have been performed.

Response: We have reanalyzed the data in Figs. 1A and 2A using ANOVA in the revised manuscript as suggested.

4. In the introduction, the authors state, “Although upregulation of Gal3 has previously been reported in microglia, the role of Gal3 in neuroinflammation is largely uncharacterized”. However, the authors then discuss 3 studies describing the role of Gal3 in neuroinflammation. This sentence should be edited.

Response: We have edited the indicated sentence in the revised manuscript to read “the role of Gal3 in inflammation is not fully understood...” (page 4, line 6).

5. In Fig 2, the authors should quantify the percentage of Gal3 positive cells that are IBA-1 positive in R6/2 mice.

Response: We have included the quantification as requested in the revised manuscript (Fig. 2E).

6. The discussion is quite lengthy and reads more like a review article. Although thorough, this should be reduced.

Response: We agree with the reviewer and have shortened the Discussion section in the revised manuscript to include only the information that is directly related to our study.

Reviewer #2

The two key weaknesses include:

1) The lack of mechanistic link between human and animal data. Increased Gal-3 in peripheral circulation is proposed as the pathophysiological mechanism in HD in humans based on the established relationship between major increase in Gal-3 levels in plasma of HD patients and disease worsening, but no studies of the role of peripheral Gal-3 were performed in mice. Thus, the link between mouse and human is missing/undeveloped;

Response: To strengthen the link between human and animal data, we have analyzed the levels of Gal3 in the plasma collected from HD mice (R6/2) and their littermate controls during disease progression. Similar to the findings in human plasma, the plasma Gal3 level was significantly up-regulated in HD mice at the disease manifested stage (12 weeks old). Given that the disease-causing transgene (the exon 1 of human *htt* gene) was driven by the human *htt* promoter, R6/2 mice exhibit many symptoms (including up-regulation of Gal3 in the brain and in peripheral circulation) parallel to HD patients. Our data showed that, in both human patients and R6/2 mice, plasma Gal3 levels might serve as a biomarker for HD progression. The cellular source(s) that contribute to the elevated plasma Gal3 and the underlying mechanism for the release of Gal3 in HD remain unclear. Given that the disease-causing transgene (exon 1 of the human *htt* gene) was driven by the human *htt* promoter, R6/2 mice exhibit many symptoms (including up-regulation of Gal3 in the brain and in peripheral circulation) parallel to HD patients. Such up-regulation of Gal3 in the plasma might be due to the expression of mHTT in blood cells of humans and mice with HD (Miller et al., Scientific Report 7: 46740, 2017; Trager et al., Brain, 137: 819- 833, 2014), which have been shown to enable blood cells (particularly the myeloid lineage cells, such as macrophages) to be more reactive for cytokine

production in an NF κ B-dependent pathway. Although various peripheral cells are able to express Gal3, macrophages have consistently been reported to be the main source of plasma Gal3 (de Boer et al., *Ann Med* 43: 60-68, 2011; de Boer et al., *J Intern Med* 272: 55-64, 2012). Therefore, it is very likely that the enhancement of plasma Gal3 in patients and mice with HD is due to the abnormally activated macrophages. We cannot exclude completely the possibility that Gal3 released by microglia in the brain might contribute to the elevation of plasma Gal3 in HD mice either. Our data show that the suppression of Gal3 in the brain of HD mice by intrastriatal delivery of lentiviruses containing shLgals3 was sufficient to reduce neuroinflammation in the brain and significantly improved major HD symptoms (e.g., motor impairment and the accumulation of mHTT inclusion, Fig. 9), demonstrating a critical role of microglial Gal3 in HD pathogenesis. The functional impact of the peripheral circulating Gal3 on HD progression is of interest and would warrant future investigation. We have included these findings and discussion in the revised manuscript (Fig. 2A; page 7, lines 5-7, page 20, lines 17-23, page 21, lines 1-13) and would like to thank the reviewer for this suggestion.

2) Multiple concerns related to insufficient characterization of microglial population and imprecision of methods used in studying isolated microglial cells.

Response: We have addressed these concerns in the revised manuscript and in the following sections as detailed below (please see comments 3, 4 & 6).

1) Among several human subjects with pathologies unrelated to HD (Suppl Table 2), three had either myocardial infarction, seizure, possible stroke, or congestive heart failure. The latter clinical conditions are known to upregulate Gal-3 expression and contribute to respective diseases.

Response: As pointed out by the reviewer, three non-HD subjects (Cases 4432, 4716, and 4631) in our study that had myocardial infarction, seizure/possible stroke, and congestive heart failure, respectively, may have higher plasma levels of Gal3 than control subjects (Edsfeldt et al., *Cerebrovasc Dis* 41: 199-203, 2016; Koukoui et al., *PLoS One* 10: e0119160, 2015). However, as discussed above, the up-regulation of Gal3 in the brain and the plasma might be two independent events. Most importantly, when compared to that of non-HD subjects (including the three abovementioned cases), the level of *LGALS3* transcript in the caudate of HD patients remained significantly higher (non-HD, 90.02 ± 19.57 , HD, 173.6 ± 20.22 , mean \pm SEM, n = 5).

2) The caudate is believed to be the susceptible region. Fig. 1G shows ~2-fold increase in *LGALS3* mRNA, whereas Suppl. Fig. 1 shows 3-6 fold increase in other brain regions in human brains. This finding should be discussed. Data in Suppl. Fig. 1 should be moved to Fig. 1.

Response: As recommended, we have moved the original Supplementary Fig. 1, which analyzed the *LGALS3* level in the cerebellum, to the revised manuscript (Fig. 1H). The data showed that there was no statistically significant difference between the HD and non-HD groups due to the variation of *LGALS3* levels in the cerebellum of HD patients. Previous studies on the impact of HD on cerebellar functions showed seemingly contradictory observations. For example, Rees and colleagues reported that although the cerebellar volume of HD patients seems smaller than that of non-HD controls, the difference did not reach statistical significance (*Mov Disord* 29: 1648-1654, 2014). Rub and colleagues, on the other hand, showed a significant loss of Purkinje cells in the cerebellum of HD patients (*Brain Pathol* 23: 165-177, 2013).

Collectively, these findings imply that the involvement of the cerebellum in HD remains elusive. Since the up-regulation of Gal3 was consistently observed in the caudate nucleus of HD patients (Fig. 1G), we chose the caudate nucleus (striatum) for further investigation. We have included this discussion in the revised manuscript (page 6, lines 17-21).

3) Vastly different morphology of microglia obtained from astrocyte-microglial cultures of WT mice shown in Fig. 2D, 3A and Suppl Fig. 3. While cells are spindle-looking in Fig. 3A, they have “activated” morphological phenotype in Fig. 2D and Suppl Fig. 3. It is unclear which panels are representative. Given the presence of ruffling in some panels, it needs to be made clear whether all experiments were conducted at a single time point. Comparative study between microglial cells from WT and R6/2 mice should be performed at least at 2 time points to rule out a possibility that it takes longer for WT microglial to acquire similar morphologic phenotype as R6/2.

Response: All comparative experiments between HD and WT groups have been carried out using microglia isolated and cultured simultaneously and under the same conditions. To rule out the possibility that WT microglia may take a longer period of time to acquire a similar phenotype as R6/2 microglia, we have performed a new experiment (Supplementary Fig. 3). Our data show that WT microglia isolated at Day in vitro (DIV21) contained less Gal3 than HD microglia isolated at DIV14 or DIV21. We have included these data in the revised manuscript (page 7, lines 18-21).

In addition, we have replaced the representative images of the original Fig. 2D and Supplementary Fig. 3 in this revised manuscript (revised Fig. 2F and Supplementary Fig. 4, respectively). We would like to thank the reviewer for pointing out this issue.

4). Proliferation rate is a plausible explanation for much higher cell density shown for microglia obtained from R6/2 mice. Gal-3 is known to affect microglial proliferation. Thus, the presence/lack of increased proliferation is critical for data interpretation in several experiments.

Response: To determine whether the proliferation rate may affect our analysis, we plated WT and HD microglia harvested at DIV21 at a density of 7,000 cells per 96-well plate for 48 hours and determined the cell numbers. Only slight reductions in the microglial numbers of both the WT and HD groups were observed, indicating that there was no proliferation of the isolated microglia under the conditions tested. In addition, no difference between the HD and WT groups was found. The cell numbers of WT and HD microglia were 6480.0 ± 243.7 and 6420.0 ± 153.0 (mean \pm SEM, Student's *t*-test, $n = 5$, $P = 0.84$), respectively. To avoid any misleading information for readers, we have replaced the representative image of the original Fig. 2D in the revised manuscript (revised Fig. 2F) and would like to thank reviewer for pointing out this matter.

5). NfκB activity is based on immunofluorescence-based appearance of p65 in the nucleus (Fig. 3), not on actual measurements of activity. Cell fractionation assay is the preferred way for evaluating protein translocation, with appropriate demonstration of cellular fraction purity. EMSA should be used to evaluate NfκB activity.

Response: To further assess NFκB activity, we have performed cell fractionation and determined the NFκB activity in nuclear extracts using an NFκB-p65 Transcription Factor Assay kit (Abcam), a nonradioactive and sensitive method for the

measurement of binding activity toward the double-stranded DNA sequence containing the NF κ B response element as an EMSA (Figs. 3C and 4C) (Wen et al., *Neuroinflammation* 15: 9, 2018; Yue et al., *Sci Rep* 5: 11149, 2015). We also used the cell fractionation method followed by western blot analyses to assess the activation of NF κ B (Supplementary Fig. 5). We would like to thank the reviewer for this suggestion.

6). **Insufficient information on the timing for experiments in isolated microglial cells.** Microglial assays are said to be run 24-48 hours after plating. The precise timing should be provided to account for possible effects of evolving cell adherence over time.

Response: As recommended, we have included the requested information in the Methods section and legends (Figs. 2, 3, 4, 5, 6) in the revised manuscript.

7) **Isolation of microglial cells directly from diseased/WT, rather than isolation from neonatal astrocyte-microglial cultures, would provide more relevant information on Gal-3 mediated effects in HD.**

Response: As recommended, we have isolated adult microglia from HD mice at the disease stage (R6/2, 12 weeks old) and their littermate controls. Similar to results obtained from neonatal microglia, adult HD microglia expressed higher levels of Gal3 and released a higher level of IL1 β and a lower level of IL10. Most importantly, treatment with a Gal3 inhibitor (TD139) significantly reduced the release of IL1 β and enhanced that of IL10. We have included these findings in the revised manuscript (Supplementary Fig. 7; page 10 lines 3-8).

In conclusion, we believe that we have performed most, if not all, of the experiments requested by both reviewers and have adequately addressed or clarified each of their points. As a result, the information in this revised manuscript has been markedly expanded, and the paper quality has also been robustly improved. It should be noted that the length and references of the revised manuscript are increased because of the addition of numerous new data and statements to conform the reviewers' requests. My colleagues and I would like to thank both reviewers again for their constructive critiques. We feel that our revised manuscript contains more than sufficient amounts of novel and important results and hope that you and the reviewers will now find our paper acceptable for publication in *Nature Communications*.

All coauthors have seen and agreed with the contents of the manuscript. None of the data contained in this report have previously been published or are under consideration elsewhere. There are no conflicts of interest. Thank you for your consideration.

Reviewers' comments:

Reviewer #1 (Remarks to the Author):

This is very nice story and sheds interesting new mechanistic light on how microglia and inflammation are modulated in neurodegenerative disease. Importantly, it shows that modulating microglial inflammatory state through a specific, potentially drugable, pathway can potently modulate disease course. While important for Huntington's disease, the focus of this paper, this could have broader implications for other neurodegenerative diseases. The authors have sufficiently addressed reviewer concerns.

Reviewer #2 (Remarks to the Author):

The revised and improved manuscript describes the role of microglial galectin-3 (Gal-3) in Huntington's disease (HD) in humans, in a mouse HD model (R6/2 mice), and in microglial cells obtained from wild type (WT) and R6/2 mice. The significance of the manuscript is in the finding that markedly increased plasma levels of Gal-3 in HD patients correlate with disease severity. The finding is novel and adds HD to the list of neurodegenerative diseases (and myocardial infarction) when increased Gal-3 in the blood correlates with severity of diseases. Using genetic tools to minimize Gal-3 upregulation, pharmacological agents that inhibit either Gal-3 or the inflammatory pathways (NfκB, NLRP3), the manuscript provides elaborate data demonstrating differences in the inflammatory signaling in microglia from HD R6/2 mouse model, predominantly NfκB pathway, and the overall Gal-3 mediated inflammatory signaling in the brain of R6/2 mice. Although the data are more comprehensive, there are continue to be a number of significant concerns, particularly in relation to potential toxicity of shGFP/shLgals3 strategy widely used in the study and a statement that Gal-3 knockdown improves clearance of damaged lysosomes.

- 1). While in humans with HD plasma Gal-3 levels are substantially higher, in R6/2 mice increase in plasma Gal-3 levels is rather modest and a link between Gal-3 appearing in the blood of HD mice and its release from microglia is weak.
- 2). Lack of morphological signs of microglial activation in R6/2 mice. Figure 2D demonstrates Gal-3 expression in brains of R6/2 mice (and not in WT mice) but all microglial cells are highly branched, consistent with morphologically quiescent cells. None of cells show morphological signs of activation. This is highly unusual and should be explained/discussed.
- 3). A statement that Gal-3 knockdown improves clearance of damaged lysosomes. The most dramatic difference in Figure 6 is the size of infected R6/2 microglia Vs. WT microglia. Visually, R6/2-shGFP and 6/2-shLgals3 cells are at least 5 times larger than respective WT cells. The cells do not seem to be swollen or dying. The abnormal morphological phenotype is seen in R6/2-shLgals3 as well. The pattern of LAMP1 is uncharacteristic and it is questionable whether strong conclusions can be made based on signal intensity changes. In addition, Figure 4 shows different sets of morphologies.
- 4). The morphology of microglial cells in brains of R6/2 mice treated with shGFP or shLgals3 virus (Figures 7 and 8) clearly shows microglial activation, contrasting that in Figure 2. But it also shows microglial activation in shGFP in WT mice (larger cell body, much thicker and shorter processes). Such changes may signify toxicity of such an approach and undermine the results, at least to some extent.
- 5). Throughout the manuscript, the authors overly rely on the signal intensity changes in their immunofluorescence studies, which is concern. Appropriate internal controls are needed or flow cytometry is to be used.

Additional concerns:

- 1). What is the meaning of “microglial inflammation” in the manuscript title? Whether the authors are trying to say that microglia mediate brain inflammation or that Gal-3 establishes a loop for microglia to further activate microglial cells, or both, the title needs to be more clear.
- 2). A statement that isolated microglial cells are pure is made based on the use of microglial marker only, which does not provide information regarding the presence of other cell types.
- 3). A statement is made that 2 mouse models of HD were used but there is only one piece of data in the Hdh-150Q model (Fig. 2I). It would be important to see the patterns of Gal-3 expression in that model.
- 4). The authors performed a requested experiment— isolate and culture microglial cells from adult WT and R6/2 mice. They show comparative data at 24 hours after plating. Such short plating time period is insufficient for microglial cells to adhere and establish processes. Also, if cellular debris is not removed it can affect the results. These two points are important given microglial phenotypes shown in R6/2 mice on Figure 2.
- 5). There is large body of data on elevated plasma Gal-3 potentially serving as biomarker in humans with myocardial infarction; the latter is relevant to some of neuropathologies and should be mentioned/discussed.

Reviewer #3 (Remarks to the Author):

The authors report a novel finding of an increase in the plasma levels of Galectin 3 in HD patients and mice. They identify that these levels are correlated with disease severity and most prominently observed in microglia. The focus on microglia is important especially given the focus on inflammation in neurodegenerative diseases in general and HD in particular. It is suggested that this upregulation of Gal3 contributes to inflammation in HD through NFkB and NLRP-3 inflammasome dependent pathway. These findings are interesting and probed very well in the manuscript. The concerns largely surround the small changes observed in lifespan expansion in the R6/2 model with a reduction of Gal3 expression. The approach to knock down the expression in vivo is specific to the striatum in the brain. It is reasonable given the critical involvement of this region in HD pathogenesis. Nonetheless, this may decrease the likelihood of observing a more extensive change in survival and body weight in this model.

This is an interesting manuscript examining the contribution of Galectin-3 to the inflammatory phenotype observed in Huntington's Disease. Overall, the authors were very responsive to important concerns raised by the previous reviewers. The experiments they performed adequately addressed issues that were related to the regulation of expression by NFkB, Gal3 puncta in lysosomes, lysosome abnormalities and HD Gal3 plasma levels.

In the abstract, the authors state “ameliorated motor dysfunction and shortened survival in HD mice.” I believe this should be lengthened or increased survival. This is a critical point. However, it must be noted that decreasing the levels of Lgals3 in the R6/2 mouse model only increased survival by about 5 days. Would this equate to any significant improvement for the patient overall? This should not be overstated.

The R6/2 is a rapidly progressing transgenic mutant huntingtin expressing mouse model. This model has such severe weight loss and early death that mechanistic studies often employ an additional HD mouse model. It is good to see the inclusion of the expression level of Lgals3 from one of the knockin mice. However, it is quite disappointing to not see any additional experiments in the knockin mice that would further strengthen the conclusions drawn using this R6/2 model.

It is important to not suggest that the R6/2 mice carry the entire promoter sequence for the human gene. Please refer to Mangiarini et al., 1996 for an accurate description of the transgene used in this model.

While the identification of abnormal "lysosome-like" structures by EM, it would have been better to perform immunoEM to more concretely identify lysosomes and the presence of Gal3 within them.

In Figure 2, it is important to be specific about which knockin mouse model was used. Please add to the Hdh designation.

The discussion still remains long. Should be shortened to focus on the critical data and conclusions observed here and decreased the reiteration of the data already presented in the Results section.

Reviewer #2

1. While in humans with HD plasma Gal-3 levels are substantially higher, in R6/2 mice increase in plasma Gal-3 levels is rather modest and a link between Gal-3 appearing in the blood of HD mice and its release from microglia is weak.

Response: The increments of Gal3 in the plasma of patients and mice with HD were very similar. Specifically, plasma Gal3 in HD subjects (3.4 ± 0.2 ng/ml, N = 26) was 18.9% higher than that of non-HD subjects (2.8 ± 0.1 ng/ml, N = 16) (Fig. 1A, P = 0.0451). Plasma Gal3 in manifest HD mice (R6/2, 12 weeks old; 19.0 ± 1.0 ng/ml, n = 11) was 21.8% higher than that of WT mice (15.6 ± 0.7 ng/ml, n = 11) (Fig. 2A, P = 0.0175). Given that HD is a degenerative disease with chronic inflammation, such a moderate increase in plasma Gal3 is expected.

Ample data from humans and mice with HD have demonstrated mHTT is expressed in both brain cells (including microglia) and peripheral cells (including macrophages). Our data showed Gal3 up-regulation in the caudate nucleus (striatum) and plasma in both human and mouse samples, suggesting that Gal3 up-regulation is an authentic HD pathogenesis. Because mHTT exists in microglia and many peripheral cells (including immune cells), the source of plasma Gal3 remains elusive and is not in the scope of this manuscript. We have included this discussion in the revised manuscript (page 19, lines 15 - 19).

2. Lack of morphological signs of microglial activation in R6/2 mice. Figure 2D demonstrates Gal-3 expression in brains of R6/2 mice (and not in WT mice) but all microglial cells are highly branched, consistent with morphologically quiescent cells. None of cells show morphological signs of activation. This is highly unusual and should be explained/discussed.

Response: We would like to thank the reviewer for bringing up this issue. In the brain of R6/2, we noticed that while some Gal3-positive microglia showed quiescent morphology, some others exhibited amoeboid morphology. Similar observations have been reported in rhesus monkey, in which Gal3-positive microglia exhibit various morphologies (e.g., ramified, amoeboid and hypertrophic) in the brain of old rhesus monkeys. Importantly, the numbers of amoeboid and hypertrophic Gal3-positive microglia are associated with cognitive impairment (Shobin et al., *Geroscience* 39: 199-220, 2017). It is possible that Gal3 up-regulation may take place before the morphological change of microglia. To avoid misleading the reader, we have replaced the original images with new images of Gal3-positive microglia of both morphologies (quiescent and amoeboid, revised Fig. 2D). We have included this discussion in the revised manuscript (page 14, lines 11 – 15).

3. A statement that Gal-3 knockdown improves clearance of damaged lysosomes. The most dramatic difference in Figure 6 is the size of infected R6/2 microglia Vs. WT microglia. Visually, R6/2-shGFP and R6/2-shLgals3 cells are at least 5 times larger than respective WT cells. The cells do not seem to be swollen or dying. The abnormal morphological phenotype is seen in R6/2-shLgals3 as well. The pattern of LAMP1 is uncharacteristic and it is questionable whether strong conclusions can be made based on signal intensity changes. In addition, Figure 4 shows different sets of morphologies.

Response: As pointed out by the reviewer, R6/2 and WT microglia exhibit various morphologies and sizes. On average, the relative sizes of R6/2 microglia were larger than WT microglia ($127 \pm 5\%$ and $100 \pm 4\%$ for R6/2 microglia ($n = 297$) and WT microglia ($n = 326$), respectively; mean \pm

SEM, Student's *t*-test, $P < 0.0001$). Nonetheless, as the reviewer suggested, these cells do not seem to be swollen or dying. We have included these data in the revised manuscript and would like to thank the reviewer for raising this issue (page 6, lines 21 – 22).

In addition to using LAMP1 as a lysosomal marker, we performed a new experiment to label lysosomes using another lysosomal marker, LAMP2, to validate the intracellular location of Gal3 puncta. Similar to our previous results with LAMP1 (Fig. 6A), we found that Gal3 puncta also colocalized with LAMP2 (revised Supplementary Fig. 7A). Because Gal3 puncta are considered a marker for damages of intracellular vesicle (Flavin et al., *Acta Neuropathol* 134: 629-653, 2017), these data further support the existence of damaged lysosomes in R6/2 microglia.

In addition to observing the changes in immunofluorescence signal intensity of LAMP1, we have quantified the amounts of LAMP1 protein by Western blot analysis and found that R6/2 microglia contained more LAMP1 proteins than WT microglia. Down-regulation of Gal3 by treatment with sh*Lgals3* significantly reduced the levels of LAMP1 (revised Supplementary Fig. 10). This finding is consistent with our hypothesis that Gal3 negatively regulates the clearance of damaged lysosomes.

4. The morphology of microglial cells in brains of R6/2 mice treated with shGFP or sh*Lgals3* virus (Figures 7 and 8) clearly shows microglial activation, contrasting that in Figure 2. But it also shows microglial activation in shGFP in WT mice (larger cell body, much thicker and shorter processes). Such changes may signify toxicity of such an approach and undermine the results, at least to some extent.

Response: We agree with the reviewer that infection with lentiviruses may facilitate microglia activation. Nonetheless, the resultant toxicity may be minimal because only very low levels of IL1 β in the infected striatal tissues were detected (Fig. 7C). No apparent activation of the NLRP3 inflammasome was found in microglia of WT mice either (Fig. 8A). The infected WT striatum exhibited no reduction in neuronal density (Fig. 9F) or reduced DARPP32 (9G, H). Overall, lentivirus infection did not significantly interfere with our assays on inflammation (Fig. 7C), neuronal survival (Fig. 9D-H), or motor performance of mice (Fig. 9A).

5. Throughout the manuscript, the authors overly rely on the signal intensity changes in their immunofluorescence studies, which is concern. Appropriate internal controls are needed or flow cytometry is to be used.

Response: Because the numbers of primary microglia that could be obtained from HD mice were limited, we had to perform most of our experiments using immunofluorescence assays. As suggested, we performed a new experiment in the revised manuscript to analyze the expression

level of Gal3 by flow cytometry and showed that R6/2 microglia expressed a significantly higher level of Gal3 compared to WT microglia (Fig. 2I). The results of flow cytometry are consistent with results of other assays (including immunofluorescence (Fig. 2F), qPCR (Fig. 2H), and Western blot (Supplementary Fig. 10)) and validate the main finding of this study that up-regulation of Gal3 occurs in HD microglia.

Additional concerns:

1) What is the meaning of “microglial inflammation” in the manuscript title? Whether the authors are trying to say that microglia mediate brain inflammation or that Gal-3 establishes a loop for microglia to further activate microglial cells, or both, the title needs to be more clear.

Response: As suggested, we have changed the title to “Galectin-3 is required for the microglia-mediated brain inflammation in Huntington’s disease”.

2) A statement that isolated microglial cells are pure is made based on the use of microglial marker only, which does not provide information regarding the presence of other cell types.

Response: For primary microglia isolated from neonatal and adult mice, we have assessed the presence of astrocytes and neurons by immunostaining assays using astrocyte-specific (GFAP) and neuron-specific (NeuN) marker. Although a small population of astrocytes (but not neurons) were found in the microglia preparations from adult mice, all Gal3-positive cells were microglia (i.e., Iba1-positive) (Supplementary Fig 6A, E, F). Among primary microglia isolated from neonatal mice, no GFAP- or NeuN-positive cells were detected (Supplementary Fig. 3B, C). This is consistent with the results from flow cytometry, which showed that 99.70 ± 0.03 % and 99.50 ± 0.05 % of WT and R6/2 microglia were CD11b-positive, respectively. CD11b is a microglial marker. These data are included in the revised manuscript (Fig. 2D, 2E; Supplementary Fig. 2A, 2B, 6A, 6B, 6E, 6F). We would like to thank the reviewer for this suggestion.

3) A statement is made that 2 mouse models of HD were used but there is only one piece of data in the Hdh-150Q model (Fig. 2I). It would be important to see the patterns of Gal-3 expression in that model.

Response: As suggested by the reviewer, we performed immunofluorescence staining of Gal3 on 21-month-old Hdh^{150Q} mice and their littermate controls, and found that Gal3 appeared in Iba1-positive, but not GFAP- or NeuN- positive, cells (Fig. 2J, Supplementary Fig. 2C, D). The results are consistent with the findings in R6/2 mice. It is interesting to note that some microglia of aged WT mice (21 months old) also expressed Gal3, but to a much lesser extent than Hdh^{150Q} mice.

Our findings with old mice are consistent with observation in old rhesus monkeys (Shobin et al., *Geroscience* 39: 199-220, 2017), suggesting that aging might contribute to the up-regulation of Gal3 in microglia of WT animals. We have included this discussion in the revised manuscript (page 14, lines 15 – 18).

4) The authors performed a requested experiment—isolate and culture microglial cells from adult WT and R6/2 mice. They show comparative data at 24 hours after plating. Such short plating time period is insufficient for microglial cells to adhere and establish processes. Also, if cellular debris is not removed it can affect the results. These two points are important given microglial phenotypes shown in R6/2 mice on Figure 2.

Response: Similar to several other studies using adult microglia, our adult microglia were used 48 hours after plating (Supplementary Fig. 6). We first allowed the microglia isolated from adult mice to sit for 24 hours after plating, then replaced the culture medium that may have contained residual debris with fresh medium containing the indicated reagent (TD139 or vehicle) for another 24 hours. Cellular debris would have been removed during the change of medium. Similar experimental designs have been utilized in other studies (e.g., Lalancette et al., *J Neurosci* 32: 10383-10395, 2012).

5) There is large body of data on elevated plasma Gal-3 potentially serving as biomarker in humans with myocardial infarction; the latter is relevant to some of neuropathologies and should be mentioned/discussed.

Response: We agree with the reviewer and have included new discussion on this part (page 18, lines 14 – 23). Specially, elevated plasma Gal3 has shown a positive correlation with cardiac infarction and systemic inflammation. Alturfan et al. has demonstrated that Gal3 as a biomarker associated with inflammation in acute myocardial infarction. (Alturfan et al., *Lab Med* 45: 336-341, 2014). Besler et al., on the other hand, reported that myocardial Gal3 is the biomarker that reflects cardiac inflammation and fibrosis, while plasma Gal3 correlates with inflammatory cell counts (Besler et al., *Circ Heart Fail* 10, 2017). Earlier studies also suggest that myocardial infarction is relevant to several types of neuropathology, such as dementia and migraine, which are tightly associated with neuroinflammation (Minett et al., 2016, *J Neuroinflam*, 13: 135-144; Ramachandran, *Semin Immunopathol.* 2018, 40: 301-314). These findings are consistent with our study because chronic inflammation is also an important pathogenesis part of HD.

Reviewer #3

1. The concerns largely surround the small changes observed in lifespan expansion in the R6/2 model with a reduction of Gal3 expression. The approach to knock down the expression in vivo is specific to the striatum in the brain. It is reasonable given the critical involvement of this region in HD pathogenesis. Nonetheless, this may decrease the likelihood of observing a more extensive change in survival and body weight in this model.

Response: As the reviewer pointed out, we specifically knocked down Gal3 in the striatum because it is a key region of HD pathogenesis. Our results allow us to explicitly analyze the role of Gal3 in striatal pathogenesis in HD. Systematic administration of blood-brain barrier (BBB)-permeable Gal3 inhibitors would be of great interest to assess their therapeutic potential, when proper Gal3 inhibitors are available in the future. We have included this information in the revised manuscript and would like to thank the reviewer for raising this issue (page 10, lines 21 – 23, page 20, lines 1 – 2).

2. In the abstract, the authors state “ameliorated motor dysfunction and shortened survival in HD mice.” I believe this should be lengthened or increased survival. This is a critical point. However, it must be noted that decreasing the levels of Lgals3 in the R6/2 mouse model only increased survival by about 5 days. Would this equate to any significant improvement for the patient overall? This should not be overstated.

Response: We agree with the reviewer. As mentioned above, the focus of this study was to explicitly analyze the role of Gal3 in striatal pathogenesis in HD. The therapeutic potential of systematic administration of BBB-permeable Gal3 inhibitors would require further investigation in the future when proper Gal3 inhibitors are available. We have included this discussion in the revised manuscript (page 20; lines 1 – 2).

3. The R6/2 is a rapidly progressing transgenic mutant huntingtin expressing mouse model. This model has such severe weight loss and early death that mechanistic studies often employ an additional HD mouse model. It is good to see the inclusion of the expression level of Lgals3 from one of the knockin mice. However, it is quite disappointing to not see any additional experiments in the knockin mice that would further strengthen the conclusions drawn using this R6/2 model.

Response: As suggested, we performed new experiments to include the expression profile of Gal3 in Hdh^{150Q} (a knock-in HD mouse model, 21 months old). Consistent with the results in R6/2 mice, Gal3 was detected in microglia (Iba1-positive), but not in astrocytes (S100) or neurons (NeuN-positive), in the striatum of Hdh^{150Q} mice (Supplementary Fig. 2C, D). Interestingly, a small portion of WT microglia in the brain of aged mice also contained Gal3, but to a much lesser extent

than Hdh^{150Q} microglia (Fig. 2J). This finding is similar to what was observed in aged WT rhesus monkeys. It is very likely that aging also contributes to Gal3 up-regulation in microglia (Shobin et al., *Geroscience* 39: 199-220, 2017). The expression levels of p65 and NLRP3 were also markedly enhanced in the Gal3-positive microglia of Hdh^{150Q} mice (Supplementary Fig. 13), as was observed in R6/2 mice. Due to the high immunostaining background caused by striatal fiber bundles in aged brains when anti-mouse 2nd antibodies were used, we were not able to quantify these images. Collectively, Gal3 upregulation and neuroinflammation were observed in both HD mouse models (R6/2, Hdh^{150Q}).

4. It is important to not suggest that the R6/2 mice carry the entire promoter sequence for the human gene. Please refer to Mangiarini et al., 1996 for an accurate description of the transgene used in this model.

Response: We have modified the indicated sentence to clarify that “the disease-causing transgene originated from the human HD gene, which included approximately 1kb of the promoter element and exon 1 of the *htt* gene...” (page 19, lines 9 – 10). We thank the reviewer for this suggestion.

5. While the identification of abnormal “lysosome-like” structures by EM, it would have been better to perform immunoEM to more concretely identify lysosomes and the presence of Gal3 within them.

Response: As suggested, we have performed immunoEM to identify lysosomes using an anti-LAMP2 antibody and Gal3 using an anti-Gal3 antibody (Supplementary Fig. 9).

6. In Figure 2, it is important to be specific about which knockin mouse model was used. Please add to the Hdh designation.

Response: We have added the Hdh^{150Q} designation to Fig. 2J, K, Supplementary Fig. 2C, D and Supplementary Fig. 13 in the revised manuscript.

7. The discussion still remains long. Should be shortened to focus on the critical data and conclusions observed here and decreased the reiteration of the data already presented in the Results section.

Response: We have shortened the Discussion in the revised manuscript as suggested.

REVIEWERS' COMMENTS:

Reviewer #1 (Remarks to the Author):

The authors have sufficiently addressed all reviewer concerns. This is an interesting and novel study, demonstrating a new mechanism by which microglia contribute to HD pathogenesis.

Reviewer #2 (Remarks to the Author):

The added several pieces of data strengthened the manuscript and addressed my previously expressed concerns.

Reviewer #3 (Remarks to the Author):

NCOMMS-18-11502B

Galectin-3 is required for the microglia-mediated brain inflammation in Huntington's disease

This is an improved manuscript from the previous submission. The authors were very responsive to my review. As stated before, this is an interesting manuscript examining the contribution of Galectin-3 to the inflammatory phenotype observed in Huntington's disease. The findings in this work will be of interest to the HD field, but also to other neurodegenerative diseases with an inflammatory component. Overall, the authors were very responsive to important concerns raised by my review. They have addressed in the text concerns that I raised concerning some interpretations of their data and overstating their data. They have added additional experiments in the second mouse model (a knockin model) expressing mutant HTT. The experiments they performed adequately addressed issues that were related to the regulation of expression by NFkB, Gal3 puncta in lysosomes, lysosome abnormalities and HD Gal3 plasma levels. They have added immunoEM to more clearly show the presence of Gal3 in the lysosomes.

They have shortened the discussion as requested. They have edited their figures to make them clearer. The inclusion of the supplementary data is very supportive of the findings presented in the main manuscript and strengthens their conclusions.

Re: Galectin-3 is required for the microglia-mediated brain inflammation in Huntington's disease (NCOMMS-18-11502C)

Our responses to referees:

Reviewer #1

The authors have sufficiently addressed all reviewer concerns. This is an interesting and novel study, demonstrating a new mechanism by which microglia contribute to HD pathogenesis.

Response: Thank you.

Reviewer #2

The added several pieces of data strengthened the manuscript and addressed my previously expressed concerns.

Response: Thank you.

Reviewer #3

This is an improved manuscript from the previous submission. The authors were very responsive to my review. As stated before, this is an interesting manuscript examining the contribution of Galectin-3 to the inflammatory phenotype observed in Huntington's disease. The findings in this work will be of interest to the HD field, but also to other neurodegenerative diseases with an inflammatory component. Overall, the authors were very responsive to important concerns raised by my review. They have addressed in the text concerns that I raised concerning some interpretations of their data and overstating their data. They have added additional experiments in the second mouse model (a knockin model) expressing mutant HTT. The experiments they performed adequately addressed issues that were related to the regulation of expression by NFkB, Gal3 puncta in lysosomes, lysosome abnormalities and HD Gal3 plasma levels. They have added immunoEM to more clearly show the presence of Gal3 in the lysosomes. They have shortened the discussion as requested. They have edited their figures to make them clearer. The inclusion of the

supplementary data is very supportive of the findings presented in the main manuscript and strengthens their conclusions.

Response: Thank you.

Sincerely,

Yijuang Chern, Ph. D.
Distinguished Research Fellow
Institute of Biomedical Sciences, Academia Sinica
Taipei, Taiwan
TEL: 886-2-26523913
Email: bmychern@ibms.sinica.edu.tw